# Ectocytosis prevents accumulation of ciliary cargo in *C. elegans* sensory neurons

Adria Razzauti, Patrick Laurent*

Laboratory of Neurophysiology, ULB Neuroscience Institute (UNI), Université Libre de Bruxelles(ULB), Brussels, Belgium

**Abstract** Cilia are sensory organelles protruding from cell surfaces. Release of extracellular vesicles (EVs) from cilia was previously observed in mammals, *Chlamydomonas*, and in male *Caenorhabditis elegans*. Using the EV marker TSP-6 (an ortholog of mammalian CD9) and other ciliary receptors, we show that EVs are formed from ciliated sensory neurons in *C. elegans* hermaphrodites. Release of EVs is observed from two ciliary locations: the cilia tip and/or periciliary membrane compartment (PCMC). Outward budding of EVs from the cilia tip leads to their release into the environment. EVs' budding from the PCMC is concomitantly phagocytosed by the associated glial cells. To maintain cilia composition, a tight regulation of cargo import and removal is achieved by the action of intra-flagellar transport (IFT). Unbalanced IFT due to cargo overexpression or mutations in the IFT machinery leads to local accumulation of ciliary proteins. Disposal of excess ciliary proteins via EVs reduces their local accumulation and exports them to the environment and/or to the glia associated to these ciliated neurons. We suggest that EV budding from cilia subcompartments acts as a safeguard mechanism to remove deleterious excess of ciliary material.

*For correspondence:
patrick.laurent@ulb.ac.be

Competing interest: The authors declare that no competing interests exist.

## Introduction

Cilia are specialized sensory compartments protruding from the cell surfaces of many cell types, including sensory neurons. Sensory cilia of the olfactory and the photoreceptors neurons concentrate the signaling components required to sense and respond to chemicals or photons, respectively. Entry and retrieval of signaling components from cilia is mediated by large intra-flagellar transport (IFT) trains moving membrane proteins bidirectionally along the ciliary microtubules. IFT operates together with cargo adapters such as the IFT-A/B and the BBSome complexes, respectively, involved in entry and removal of transmembrane proteins from cilia. Defects in cilia trafficking can alter ciliogenesis, cilia structure, and composition and ultimately cilia signaling (*Fliegauf et al., 2007*; *Nachury and Mick, 2019*). *Caenorhabditis elegans* proved to be an excellent experimental system to identify and analyze the genes required for cilia function (*Scholey et al., 2004*). Sensory cilia integrity can be evaluated by sensory responses of the animals or by neuronal uptake of lipophilic dye (DiI) via its exposed ciliated ends (*Perkins et al., 1986*; *Starich et al., 1995*). Mutations reducing cilia length cause dye filling defective phenotypes (Dyf) together with sensory defects (*Inglis et al., 2007*).

Sensory organs combining glial-like cells and ciliated sensory neurons are observed across metazoans, ranging from the invertebrate sensilla to the mammalian olfactory epithelium (*Tang et al., 2020*; *Wright, 1992*). In these sensory organs, glia contributes to sensory function by releasing trophic factors, recycling neurotransmitters, controlling ion balance, and pruning synapses (*Fredieu and Mahowald, 1989*; *Oikonomou and Shaham, 2011*; *Vecino et al., 2016*). In *C. elegans*, all neuroepithelial glial cells associate with the ciliated end of sensory neurons and together form sensilla. These include the deirids, outer labial, inner labial, cephalic, phasmid, and amphid sensilla. The amphid sensilla is the primary sensory organ of *C. elegans* allowing to sense external cues. The amphids are

bilateral sensory organs, each formed by 12 ciliated neurons and 2 glial cells creating an epidermlike continuum with the hypoderm. Two glial cells, called AMphid sheath (AMsh) and AMphid socket (AMso), delimit together a matrix-filled pore opened to the external environment that houses the nerve receptive endings (NREs) of 8 of the 12 amphid neurons. The NREs of the other four amphid neurons (AWA, AWB, AWC, AFD) are fully embedded within AMsh (*Figure 1A and B*).

Extracellular vesicles (EVs) are membrane-limited vesicles released by cells. EVs hold exciting significance for biology, pathology, diagnostics, and therapeutics (*van Niel et al., 2018*). EVs are heterogeneous vesicles, including the <150 nm diameter exosomes derived from multivesicular bodies (MVBs) and the usually larger (100 nm to 1 μm) ectosomes formed via outward budding of the plasma membrane (*Colombo et al., 2014*; *van Niel et al., 2018*). The capability for cilia to release ectosomes into the extracellular space has been described in a variety of organisms including *Chlamydomonas* (*McLean and Brown, 1974*; *Bergman et al., 1975*; *Wood et al., 2013*), *C. elegans* (*Wang et al., 2014*), and mammals (*Nager et al., 2017*). Analysis of mammalian and nematode EVs content has detected several proteins that were originally enriched in the cilia, in particular polycystic kidney disease (PKD) protein polycystin-2/PKD2 (*Wood and Rosenbaum, 2015*).

In *C. elegans*, the polycystin-2 ortholog PKD-2 localizes to the cilia of a subset of male ciliated sensory neurons (*Barr et al., 2001*). In overexpression strains, PKD-2-GFP containing EVs are released from male ciliated neurons into the external environment and EVs purified from male contribute to inter-individual communication (*Wang et al., 2014*; *Wang et al., 2020*). Electron microscopy of the male cephalic sensilla revealed EVs accumulating in the lumen surroundings the periciliary membrane compartment (PCMC) of male cephalic neurons (CEM neurons), hinting for a physiological release from CEM PCMCs (*Wang et al., 2014*). Multiple evidence suggests that ciliary EVs correspond to ectosomes shed from the plasma membrane of the males ciliated neurons rather than exosomes derived from MVBs. First, the production of ciliary EVs was not affected by mutants disrupting MVBs maturation, including mutants disturbing the ESCRT complex *stam-1*, *mvb-12,* and *alx-1*. Second, MVBs were not observed in cilia or in the distal dendrite. Finally, some omega-shaped structures were observed at the PCMC of CEM neurons (*Wang et al., 2014*; *Silva et al., 2017*).

Several questions remain unanswered: where and how EVs bud from cilia in physiological or pathological conditions, how cargoes enter ciliary EVs, and what is the physiological function of these ciliary EVs. Here, we show that most ciliated sensory neurons of *C. elegans* can pack and export ciliary membrane proteins in two types of ciliary ectosomes. Ectosomes formed at the tip of the cilia, which are released to the environment, and ectosomes formed at the base of the cilia (PCMC), which are readily phagocytosed by the surrounding glial cells. Defects in cilia entry of the salt-sensing guanylate cyclase GCY-22 in mutants for an anterograde motor kinesin (*osm-3*) give rise to basal ectosomes budding from the PCMC. Conversely, mutations in the BBsome (Bardet–Biedl syndrome) protein BBS-8 results in defective receptor retrieval and an enhanced budding from cilia tip. Mutants for the clathrin adaptor AP-1 complex subunit mu-1 (*unc-101*) reduce sorting of GCY-22 receptor to cilia and abolish ciliary ectosome production. Therefore, trafficking bias facilitates cargo entry into apical or basal ectosomes, according to cargo accumulation in cilia or PCMC, respectively. In addition, cargoes might contribute themselves to membrane bending or to their sorting to bending membranes. We suggest that ectocytosis might contribute to reduce pathological accumulations of ciliary proteins. Although ectocytosis is maintained in the absence of glia, we observe that basal ectosome biogenesis is coupled to ectosome removal by phagocytic glia. This neuron-glia coupling maintains the cilia structure and sensory function of a subset of ciliated neurons.

## Results

### Ciliated neurons transfer DiI-stained membrane to its neighboring glia in an ATP-dependent manner

We used the amphid sensilla as a model of anatomically connected neurons and glia. When animals are soaked in the lipophilic dye DiI, a subset of environmentally exposed amphid sensory neurons (and non-exposed AWB neuron) uptake DiI from their sensory cilia (*Perkins et al., 1986*; *Inglis et al., 2007*; *Figure 1A and B*). Although not being environmentally exposed, the AMsh glial cells embed the PCMC of the DiI-stained neuronal subset. Interestingly, when staining the animals with DiI, AMsh glia shows a puncta-like staining pattern, in contrast to the strong homogenous staining observed in

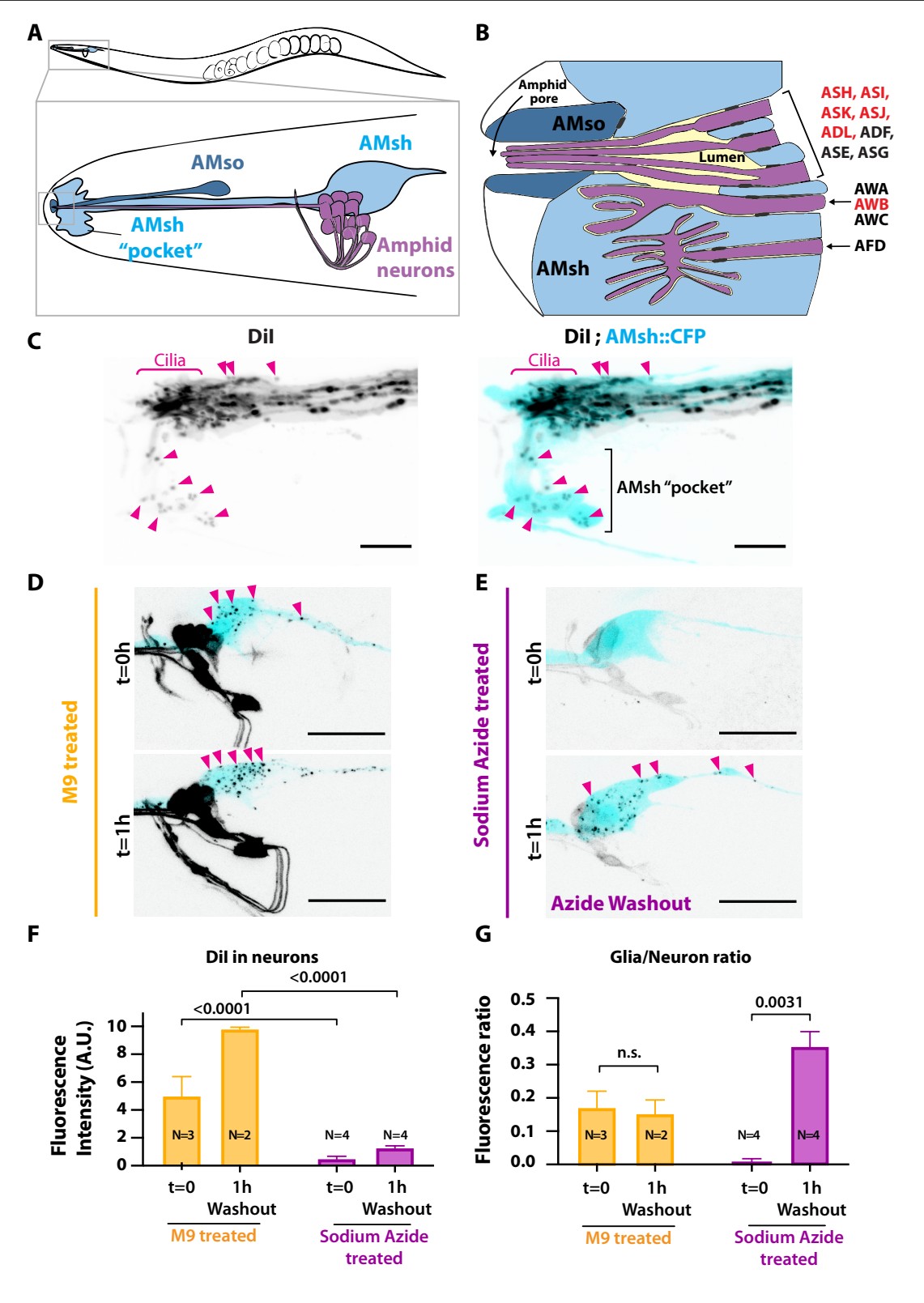

**Figure 1.** Ciliated amphid neurons transfer DiI-stained membrane to the ensheathing glia in an ATP-dependent manner within minutes. (**A**) Anatomical organization of the amphid sensilla in *C. elegans* (top). Head close-up scheme shows AMsh and AMsh 'pocket' (light blue), AMso (dark blue), and the amphid neurons (magenta). (**B**) Schematic depicting the nerve receptive ending (NRE) of the 12 amphid neurons. Tight junctions between neurons and glia, or between AMsh and AMso, are depicted as dark gray discs between cells. The red labeled neurons are the DiI-stained subset. (**C**) Maximum

*Figure 1 continued on next page*

*Figure 1 continued*

intensity projection of the DiI-stained neurons in a strain expressing CFP in AMsh glia. Neuronally derived vesicles containing DiI can be observed within the AMsh cytoplasm and in the AMsh 'pocket' (magenta arrowheads) (**D**) DiI staining of the amphid neurons treated with M9 at t = 0 and after 1 hr of being washed with M9, amphid neurons strongly stained, and multiple vesicular puncta within AMsh cell body can be observed (magenta arrowheads). (**E**) In the presence of 25 mM sodium azide, DiI staining of the amphid neurons is fainter and no vesicles could be observed within AMsh cell body at t = 0. After 1 hr of being washed with M9 to remove sodium azide (azide washout), the AMsh staining is recovered. (**F**) Measurements of fluorescence intensity (in arbitrary units) quantified in neuronal cell bodies. Two-way ANOVA, Sidak's correction for multiple comparison. (**G**) Glia/neuron fluorescence ratios. Glia fluorescence was normalized to the fluorescence intensity of neurons. AMsh normalized fluorescence is drastically reduced in the presence of sodium azide and increases after its removal (t = 1 hr). Unpaired t-test with Welch's correction. Scale bars: 5 µm in (**C**), 20 µm in (**D, E**).

The online version of this article includes the following figure supplement(s) for figure 1:

**Figure supplement 1.** Experimental design for DiI experiment.

neurons. Interestingly, this staining of the AMsh glia only occurs if the neurons themselves are priorly stained (*Ohkura and Bürglin, 2011*). Together, these observations imply that DiI is first incorporated in membranes of the amphid neurons and secondarily these DiI-stained neuronal membranes are exported to AMsh.

Whether, where, and how ciliated neurons export DiI-stained membrane to the glial cell remains unanswered. To address these questions, we analyzed the DiI filling dynamics in the amphid sensilla. DiI accumulation in neurons and in glia was rapid: soaking the animals in DiI for 20 min was enough to stain amphid and phasmid neurons as well as the amphid and phasmid sheath glia (*Figure 1C* and *Figure 1—figure supplement 1B*). High-resolution images of the nose tip revealed DiI-stained vesicles within AMsh cytoplasm, either next to the dye-filled cilia or further away, accumulating in a region of AMsh that we hereafter named as the AMsh 'pocket' (*Figure 1C*). Interestingly, time-lapse imaging showed that these DiI-stained vesicles were produced where neuron ciliated endings are located. Once released, the flow of these vesicles was directed towards AMsh 'pocket' or towards AMsh cell body (*Video 1*). Finally, DiI-containing vesicles were seen to accumulate in the soma of AMsh (*Figure 1D*).

Sodium azide (NaN₃) uncouples oxidative phosphorylation and rapidly depletes intracellular ATP, disturbing membrane trafficking (*Prescianotto-Baschong and Riezman, 1998*). ATP-dependent membrane trafficking events are required for the fast DiI staining of the neurons. When worms are treated with 25 mM sodium azide 15 min prior and during DiI staining, the neuron and glia staining entirely disappeared (*Figure 1—figure supplement 1C*). However, when the animals are directly soaked for 20 min in a solution of DiI supplemented with 25 mM sodium azide, we observe a faint staining of the neurons while AMsh glia remained unstained (*Figure 1E*, t = 0 hr; see *Figure 1—figure supplement 1A* for experimental scheme). This observation suggests that amphid neurons captured DiI but did not transfer it to AMsh in the absence of ATP. After 1 hr recovery in the absence of sodium azide (washout condition), the staining of AMsh was retrieved, suggesting that the ATP-dependent transfer of DiI restarted (*Figure 1E*, t = 1 hr). When normalizing the AMsh staining to the neuronal fluorescence, DiI transfer from neurons to glia was re-established after washout (*Figure 1F and G*).

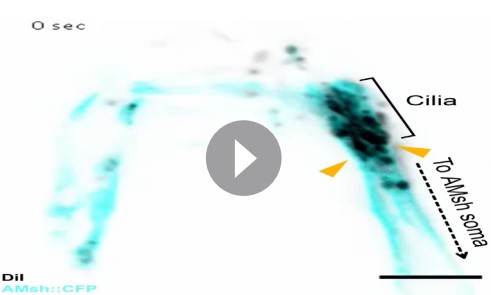

**Video 1.** Extracellular vesicles (EVs) are released from the DiI-uptaking amphid neurons and are captured by the surrounding glia. An animal expressing AMsh cytoplasmic CFP and stained with DiI was immobilized with 10 mM tetramisole and recorded during approximately 8 min. Membrane fragments become detached from ciliary regions; these DiI-carrying vesicles are sequentially trapped by glial cells (orange arrows). Dashed arrow indicates directional flow of vesicles towards AMsh cell body. AMsh 'pocket' is indicated and contains multiple already-exported vesicles. Scale bar: 5 µm.

https://elifesciences.org/articles/67670/figures#video1

## The tetraspanins TSP-6 and TSP-7 enter ciliary EVs captured by the surrounding glial cells

We hypothesized that ciliary EVs released from the amphid neurons mediate the DiI export to AMsh. This scenario implies that EV markers should also be exported from amphid neurons to AMsh. To our knowledge, all current EV protein

markers described for *C. elegans* are proteins expressed in male-specific ciliated sensory neurons and in sex-shared IL2 neurons (*Wang et al., 2014*; *Maguire et al., 2015*). To label ciliary EVs and show their export to AMsh, we explored other potential EV markers. The tetraspanin CD63, CD9, and CD81 are commonly used as EV markers in mammals, and ciliary localization of CD9 and CD81 was previously observed in IMCD3 cells (*Kim, 2013*; *Mick et al., 2015*; *Keerthikumar et al., 2016*). *C. elegans* bears 20 tetraspanin genes (TSP-1–20). Using a reciprocal best hit approach, TSP-6 and TSP-7 appeared orthologous to CD9 and CD63, respectively (*Figure 2A*; *Hemler, 2005*; *Moreno-Hagelsieb and Latimer, 2008*; *Pir et al., 2021*). As potential ciliary EVs markers for the amphid neurons, we selected TSP-6, which is strongly expressed in the ciliated neurons, and TSP-7, which is broadly expressed in the nervous system but not in ciliated neurons (*Hammarlund, 2018*; *Lorenzo et al., 2020*; *Figure 2A*). However, a valid ciliary EV marker should have two properties: (1) it should be enriched in cilia, and (2) it should be loaded as a cargo in EVs released from these cilia.

First, to answer if TSP-6 and TSP-7 were enriched in the cilia, we developed TSP-6-wrmScarlet and TSP-7-wrmScarlet fluorescent fusion proteins and generated transgenic animals overexpressing these constructs in specific neurons or subsets of neurons. Each neuron or group of neurons was chosen for the positioning of its NREs with respect to their supporting glia. In each of the neurons we examined – including the ASER, AFD, ASH, ASI, ADL, ADF, AWA, and IL2 neurons – TSP-6-wrmScarlet and/or TSP-7-wrmScarlet appeared enriched at the apical NREs (*Figure 2* and *Figure 2—figure supplements 1 and 2*). The canonical sensory cilium is subdivided in various sub-ciliary domains (*Figure 2B*). Quantification showed that TSP-6-wrmScarlet was enriched at the ASER cilium and PCMC compared to the distal dendrite (*Figure 2C' and C''*). Similarly, using the dynein light intermediate chain XBX-1 marker to label the axoneme and PCMC, we saw that XBX-1-mEGFP largely overlapped with TSP-7-wrmScarlet location in ASH, ADL, ADF cilia, further supporting its enrichment in cilia and PCMC (*Figure 2—figure supplement 1B*).

Second, if TSP-6-wrmScarlet was loaded in EVs released by amphid neuron cilia, we should observe TSP-6-wrmScarlet in EVs located either in the amphid pore or exported to/captured by AMsh, as we observed for DiI. Indeed, when overexpressed from a subset of amphid neurons, these markers were exported from their cilia to EVs observed in the amphid pore and/or in AMsh cytoplasm. Within AMsh, TSP-6-wrmScarlet and TSP-7-wrmScarlet were always observed as intracellular vesicles (*Figure 2C–F*, *Figure 2—figure supplement 1A, B', C'*, and *Figure 2—figure supplement 2B*). However, because these vesicles originate from the neurons, we will call them EVs, something that is justified later. When expressed in the right ASE neuron (ASER), TSP-6-wrmScarlet and TSP-7-wrmScarlet were exported to fluorescent EVs located only in the right AMsh cell (AMshR) (*Figure 2C* and *Figure 2—figure supplement 1A*). We did not observe EVs in AMso glial cells nor in the contralateral AMshL, suggesting that EV export occurs from the ASER PCMC, the only place where ASER directly contacts AMshR. Accordingly, we observed TSP-6-wrmScarlet-carrying EVs in close proximity to the ASER PCMC as well as in the AMsh 'pocket' (*Figure 2C*, insets, magenta arrowheads). We also observed fluorescent EVs in the amphid pore, suggesting that these can also be apically released from ASER cilium into the amphid pore (*Figure 2C*, insets, green arrowheads). Similarly, we observed export of TSP-6-wrmScarlet-carrying EVs from the bilateral ASH and ASI neurons to the amphid pore and to both AMsh cells (and rarely in AMso glia; *Figure 2D*). We observed export of TSP-7-wrmScarlet-carrying EVs from the bilateral ASH, ADL, ADF, AWA neurons to both AMsh cells (*Figure 2—figure supplement 1B*). Finally, we assayed the AFD neurons as their NREs consist of many microvilli and a single cilium all embedded within AMsh cytoplasm. When expressed in AFD neurons, TSP-6-wrmScarlet and TSP-7-wrmScarlet were observed at the surface of microvilli and AFD base (*Figure 2E* and *Figure 2—figure supplement 1B*). High-resolution imaging allowed us to see TSP-7-wrmScarlet in a vesicular compartment within AFD base, possibly corresponding to a recycling or trafficking compartment (*Figure 2—figure supplement 2A*). Similar export properties were observed for AFD neurons: EVs carrying TSP-6-wrmScarlet or TSP-7-wrmScarlet were exported from bilateral AFD neurons to both AMsh glial cells (*Figure 2E* and *Figure 2—figure supplement 2B*). Fluorescent EVs were observed within the cytoplasm of AMsh glia, around the microvilli of AFD and also in the AMsh 'pocket' area. TSP-positive EVs were also observed in AMsh cell body (*Figure 2E* and *Figure 2—figure supplement 2A and B*). Export of EVs carrying TSP-7-wrmScarlet from AFD neurons to AMsh was observed across all larval stages from L1 stage to adult. As the animals aged, we observed a progressive build-up of TSP-7-wrmScarlet EVs in AMsh glia (*Figure 2—figure supplement 2B*), suggesting that this export of TSP-7-wrmScarlet EVs occurs continuously from AFD neurons to AMsh and starts at early larval stages.

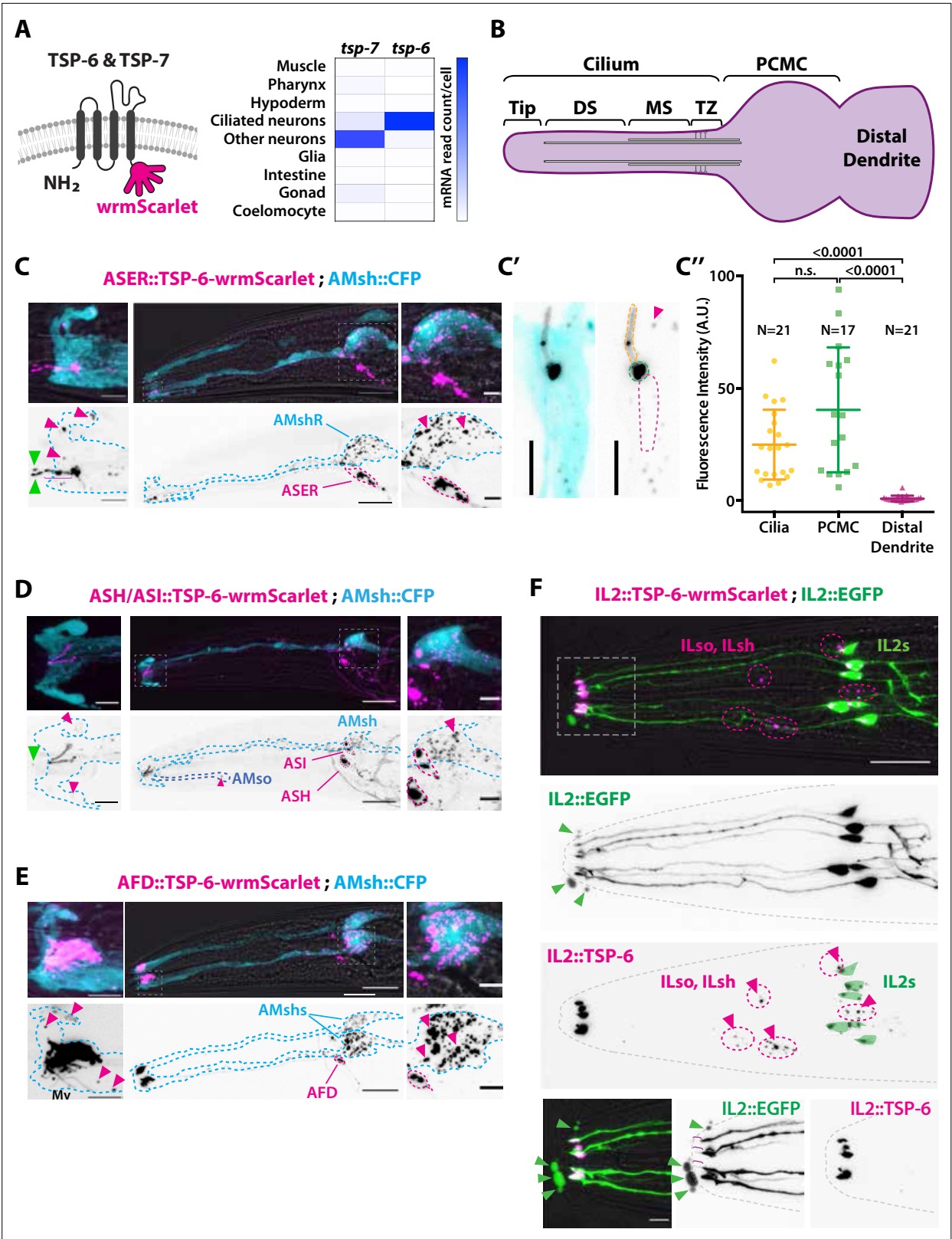

**Figure 2.** The extracellular vesicle (EV) marker TSP-6-wrmScarlet localizes to the cilia region, is loaded into EVs, and exported to surrounding glial cells. (**A**) TSP-6 and TSP-7 were C-terminally tagged with wrmScarlet. Expression patterns based on molecular cell profiling show that *tsp-6* and *tsp-7* genes are enriched in ciliated or non-ciliated neurons, respectively. (**B**) Scheme of ciliary domains: ciliary tip, distal segment (DS), middle segment (MS), transition zone (TZ), and the periciliary membrane compartment (PCMC) located at the base of the cilia in contact with the distal dendrite.

*Figure 2 continued on next page*

*Figure 2 continued*

(**C**) Expression of TSP-6-wrmScarlet in ASER driven by *gcy-5* promoter. TSP-6-wrmScarlet-carrying EVs exported from ASER are observed within the cytoplasm of AMsh. Left panels show a magnification of ASER cilium region, EVs are released in the cilia pore anterior to the ASER cilia tip (green arrowhead). Right panels show EVs within AMsh cell body (magenta arrowheads). Scale bar: 20 μm middle panel and 5 μm insets. (**C'**) TSP-6-wrmScarlet is enriched in ASER cilium: representative confocal projection showing TSP-6-wrmScarlet enrichment in PCMC and cilium. (**C''**) Fluorescence quantification in animals expressing TSP-6-wrmScarlet in ASER neuron. Brown–Forsythe ANOVA, multiple comparisons corrected by Dunnett´s test. (**D**) Expression of TSP-6-wrmScarlet under the *sra-6* promoter (driving expression in ASH and ASI neurons). TSP-6-wrmScarlet is enriched in both ASH and ASI cilia. Left panel shows TSP-6-wrmScarlet-carrying EVs released by ASI and/or ASH in the cilia pore (green arrowhead), EVs released by ASH/ASI were also seen within the cytoplasm of AMsh surrounding ASI/ASH cilia (magenta arrowheads). Few vesicles were also observed in AMso (blue dashed outline, middle panel). Right panel shows AMsh soma with multiple EVs (magenta arrowhead). Scale bar: 20 μm middle panel and 5 μm insets. (**E**) Expression of TSP-6-wrmScarlet in AFD neurons driven by *gcy-8* promoter. TSP-6-wrmScarlet is enriched in AFD microvilli and PCMC. Left panel shows TSP-6-wrmScarlet-carrying EVs within the cytoplasm of AMsh that surrounds AFD terminals (magenta arrowheads). Right panel shows AMsh soma with multiple EVs (magenta arrowhead). Scale bar: 20 μm middle panel and 5 μm insets. (**F**) Co-expression of TSP-6-wrmScarlet and cytoplasmic mEGFP in IL2 neurons (driven by *klp-6* promoter). mEGFP can be observed within EVs that are environmentally released (green arrowhead) while TSP-6-wrmScarlet is observed on EVs located within the cytoplasm of ILsh and ILso glial cells (magenta arrowheads). Theoretical position of ILsh and ILso was outlined (magenta dashed circles), IL2 neurons position was drawn with green filled outlines. Scale bar: 20 μm top panels and 5 μm insets.

The online version of this article includes the following figure supplement(s) for figure 2:

**Figure supplement 1.** TSP-7-wrmScarlet also localizes to the cilia region, is loaded into extracellular vesicles (EVs), and exported to surrounding glial cells.

**Figure supplement 2.** TSP-7-wrmScarlet localizes to AFD nerve receptive endings (NREs), is loaded into extracellular vesicles (EVs), and exported to embedding AMsh glia.

---

Therefore, we show that TSP-6-wrmScarlet and TSP-7-wrmScarlet label ciliary membrane compartments. These markers can be loaded into EVs that are released from amphid neurons to amphid pore and/or to AMsh.

## Ectosome biogenesis from two ciliary locations produces divergent EVs' fates

The cilia of male IL2 neurons were previously described to produce and release EVs to the environment (*Wang et al., 2014*). Accordingly, we could also observe EV release occurring from the hermaphrodite IL2 cilia expressing different markers: TSP-6-wrmScarlet co-expressed with cytoplasmic mEGFP, TSP-7-wrmScarlet co-expressed with cytoplasmic mEGFP or cytoplasmic mCherry alone. Expression of TSP-7-wrmScarlet and TSP-6-wrmScarlet appeared enriched at PCMC of IL2s compared cilia proper, while cytoplasmic mEGFP or mCherry were homogenously distributed in IL2s cilia (*Figure 2F* and *Figure 2—figure supplement 2C*). The EVs released outside the animals were always carrying cytoplasmic mEGFP or mCherry (*Figure 2F* and *Figure 2—figure supplement 2D*) but were only occasionally carrying TSP-7-wrmScarlet and TSP-6-wrmScarlet. In contrast, EVs exported from IL2 neurons to their supporting glia were always carrying TSP-6-wrmScarlet and TSP-7-wrmScarlet (*Figure 2F* and *Figure 2—figure supplement 2C'*). The stained glia presumably corresponded to the inner labial sheath (ILsh) and socket (ILso) glial cells, based on their location. Therefore, IL2 neurons likely release two types of EVs, EVs transferred to the glia carrying TSP-6-wrmScarlet and TSP-7-wrmScarlet, and EV released outside the animals that often lack the tetraspanin markers.

To analyze the dynamic of biogenesis, release and transfer of ciliary EV, we performed live imaging of animals co-expressing the cytoplasmic mEGFP along with TSP-7-wrmScarlet/TSP-6-wrmScarlet in IL2 neurons. These animals were immobilized with 10 mM tetramisole, an agonist of cholinergic receptors that force muscular paralysis

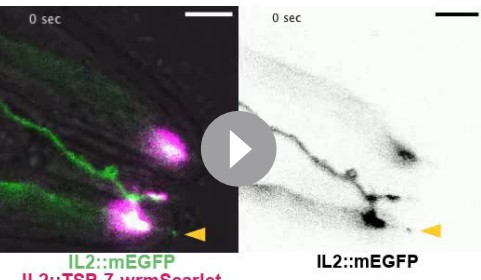

**Video 2.** Apical ectosomes are released from the cilia tip of IL2. An animal expressing cytoplasmic mEGFP and TSP-7-wrmScarlet in IL2 neurons was immobilized with 10 mM tetramisole and recorded during approximately 3 min. Two ectosomes grow from the cilia tip (yellow arrowheads), displayed ectosomes carry only mEGFP. Scission events occur at t = 160 s and t = 167 s. Scale bar: 5 μm.
https://elifesciences.org/articles/67670/figures#video2

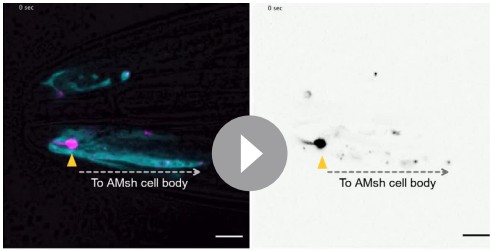

**Video 3.** Apical ectosomes are released from the cilia tip of IL2. An animal expressing cytoplasmic mEGFP and TSP-7-wrmScarlet in IL2 was immobilized with 10 mM tetramisole and recorded during approximately 3 min. Dynamics of a large distal ectosome (yellow arrow) growing from IL2 cilia tip carrying both mEGFP and TSP-7-wrmScarlet until the scission event occurs (at t = 164 s). Scale bar: 5 µm.
https://elifesciences.org/articles/67670/figures#video3

pGCY-5::TSP-6::wrmScarlet

**Video 5.** Basal ectosomes are released from the periciliary membrane compartment (PCMC) of ASER. An animal expressing TSP-6-wrmScarlet in ASER was immobilized with 10 mM tetramisole and recorded for 365 s. Small-sized ectosomes carrying TSP-6-wrmScarlet are released from ASER PCMC (yellow arrowhead), the released ectosomes are directed towards AMsh cell body (blue, CFP cytoplasmic expression, dashed arrow indicates directionality towards AMsh cell body). Scale bar: 5 µm.
https://elifesciences.org/articles/67670/figures#video5

but do not interrupt ATP-dependent processes (*Reilly et al., 2017Reilly et al., 2017*). In these conditions, we observed outward budding of the plasma membrane typical of ectosomes in IL2 neurons. The events occurred from two ciliary locations: first, ectosomes varying from below 250 nm to ~2 µm diameter are formed and released from the IL2 cilia tip by a constant flow of membrane towards a cilia protrusion at IL2 cilia tip. Scission of the ectosomes from the cilia tip seemed to occur randomly, releasing ectosomes of variable size in these preparations. These distal ectosomes did not always carry the TSP-7-wrmScarlet or TSP-6-wrmScarlet markers (*Video 2*). TSP-7-wrmScarlet appeared sorted or not into the IL2 apical ectosomes according to its presence or absence at the cilia tip at the time of ectosome budding. An ~2 µm diameter distal ectosome was formed and released in ~150 s, suggesting a fast directional membrane flow towards the cilia tip of IL2 neurons (*Video 3*). Second, IL2 neurons also released ectosomes carrying TSP-7-wrmScarlet from their PCMC. These were observed as translocating from IL2 PCMC towards ILsh or ILso cytoplasm (*Video 4*). We also performed in vivo live recordings from animals expressing TSP-6-wrmScarlet in ASER neurons. We observed a similar flow of barely detectable basal ectosomes released from ASER PCMC, suggesting local production of ectosomes (*Video 5*).

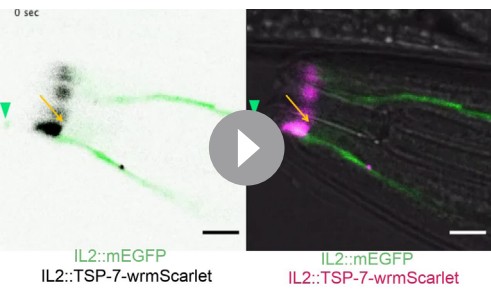

**Video 4.** Basal ectosomes are released from the periciliary membrane compartment (PCMC) of IL2s. An animal expressing cytoplasmic mEGFP and TSP-7-wrmScarlet in IL2 neurons was immobilized with 10 mM tetramisole and recorded during 101 s. Basal ectosomes (yellow arrows) are released and flow towards the IL2 sheath/socket glia cell bodies. An already released apical ectosome can be observed attached to the animal's nose (green arrowhead). Scale bar: 5 µm.
https://elifesciences.org/articles/67670/figures#video4

Altogether, our results suggest a model where ciliary ectosomes mediate the export of ciliary membrane containing either: DiI, membrane proteins like tetraspanins (TSP-6-wrmScarlet or TSP-7-wrmScarlet) and/or ciliary cytoplasm (including mEGFP or mCherry). Ectosome biogenesis up to scission is fast but variable ranging from few seconds to few minutes, leading to variable ectosome size. These ciliary ectosomes can bud from the NRE of multiple ciliated neurons, either from the PCMC and/or from cilia tip. Markers enriched at PCMC are also more likely to enter in basal ectosomes than markers enriched in distal cilia. When budding from PCMC, ectosomes are subsequently captured by the contacting glial cells. When ectosomes are budding from ciliary tip, these are not captured by the adjacent glial cells but instead they are environmentally released.

## Ciliary receptors can be sorted to ectosomes and exported to AMsh

Previous work showed that only a subset of overexpressed ciliary proteins enters *C. elegans* male EVs (*Maguire et al., 2015*; *Wang et al., 2020*). To explore which cilia proteins can enter basal ectosomes, we overexpressed fluorescently tagged ciliary membrane proteins known to localize to the cilia of AFD, ASER, and ASK neurons. The transmembrane guanylyl cyclase receptor GCY-8 is localized to AFD receptive endings where it contributes to the thermotaxis behavior of *C. elegans* (*Inada et al., 2006*). Overexpression of the fusion protein GCY-8-wrmScarlet in AFD showed enrichment in all the distal AFD NREs, including microvilli and PCMC. GCY-8-wrmScarlet was exported from AFD to AMsh, producing fluorescent EVs that surrounded AFD microvilli and that were ultimately trafficked to AMsh cell body (*Figure 3A*). We also tagged a G-protein coupled receptor (GPCR) endogenously localized in AFD microvilli, the SRTX-1 receptor (*Nguyen et al., 2014*). When overexpressed in AFD neurons, SRTX-1-wrmScarlet was also exported from AFD terminals to EVs in AMsh glia (*Figure 3B*). We next tagged the SRBC-64 GPCR involved in pheromone sensing and known to localize to the cilia and PCMC of the ASK neurons (*Kim, 2009*). Overexpression of SRBC-64-wrmScarlet in ASK neurons localized to the cilia and PCMC but was not exported to AMsh (*Figure 3C*). Finally, we tagged the transmembrane receptor guanylyl cyclase GCY-22, a specific ASER cilium-located receptor involved in salt sensing (*Ortiz et al., 2009*). The overexpressed GCY-22-wrmScarlet fusion protein localized in a bi-partite distribution at the cilium tip and PCMC of ASER, as previously described (*van der Burght et al., 2020*; *Figure 3D*). We observed that GCY-22-wrmScarlet was also exported from the PCMC of ASER to AMsh, producing fewer but noticeably larger (~1 µm) fluorescent EVs in AMsh cell body (*Figure 3D*). In addition, GCY-22-wrmScarlet was observed enriched at the membrane of large protrusions budding from ASER PCMC (*Figure 3D*, orange arrowhead). Because of their large size, budding of GCY-22-wrmScarlet-carrying EVs could be resolved, and their budding dynamics recorded in vivo. Multiple recorded videos revealed the buddings of ~1 µm diameter ectosomes, starting with the elongation of a protrusion from ASER PCMC followed by the narrowing of the tubule connecting the protrusion to the PCMC up to the scission event (*Video 6*). All budding events took place within 6–25 min of recording (N = 3). The entire budding, neck elongation, and scission of basal ectosomes containing GCY-22 happened in close contact with AMsh, hinting to a phagocytic process. As soon as scission occurred, the large vesicles moved retrogradely towards AMsh cell body. Therefore, most of the ciliary membrane proteins we assessed can be loaded into ciliary EVs when overexpressed. However, there are exceptions exemplified by SRBC-64-wrmScarlet, suggesting that some ciliary membrane proteins do not enter EVs or their levels are below the limits of detection in the used setup.

## Apical or basal ectocytosis prevents accumulations of cargo in cilia trafficking mutants

Previous experiments showed that large ectosomes carrying GCY-22-wrmScarlet were formed at the PCMC of ASER and simultaneously captured by AMsh when overexpressed. Assuming that the number of GCY-22-wrmScarlet vesicles in AMsh cell body reflects the strength of the GCY-22-wrmScarlet export by ectocytosis, these transgenic animals were used as a tool to quantify EV biogenesis from ASER. GCY-22-wrmScarlet transgenics were crossed into mutants modulating the amount of GCY-22-wrmScarlet at cilia. We first asked if ectocytosis required a functional cilium to occur or can occur simply by virtue of GCY-22-wrmScarlet accumulation at distal dendrite. In *C. elegans*, most genes involved in ciliogenesis are under the transcriptional control of the RFX-type transcription factor DAF-19 (*Swoboda et al., 2000*). *daf-19(m86)* mutant worms are completely void of ciliated structures. Instead amphid dendrites terminate in club-shaped ectopic membrane and form junctions with AMsh. Importantly, traffic to the distal dendrite of several cilia proteins is maintained in *daf-19* (*Dwyer et al., 2001*). In *daf-19(m86)* mutants, GCY-22-wrmScarlet accumulated in an elongated ectopic membrane compartment protruding from the distal dendrite in close contact with AMsh (*Figure 4A*). The export of GCY-22-wrmScarlet from ASER to AMsh still occurred from this ectopic compartment and the number of GCY-22-wrmScarlet vesicles in AMsh was increased respective to wild-type controls (*Figure 4B*), suggesting that the cilia structure itself is not necessary for ectocytosis to AMsh to occur. We next asked whether a strong traffic of GCY-22-wrmScarlet towards the cilia is required for its ectocytosis. The AP-1 µ1 clathrin adaptor UNC-101 is not required for cilia formation but acts in the *trans*-Golgi network and endosomes to mediate ciliary protein sorting and trafficking towards the cilia (*Dwyer et al., 2001*). In *unc-101(m1)* mutants, GCY-22-wrmScarlet evenly localized in distal dendrite

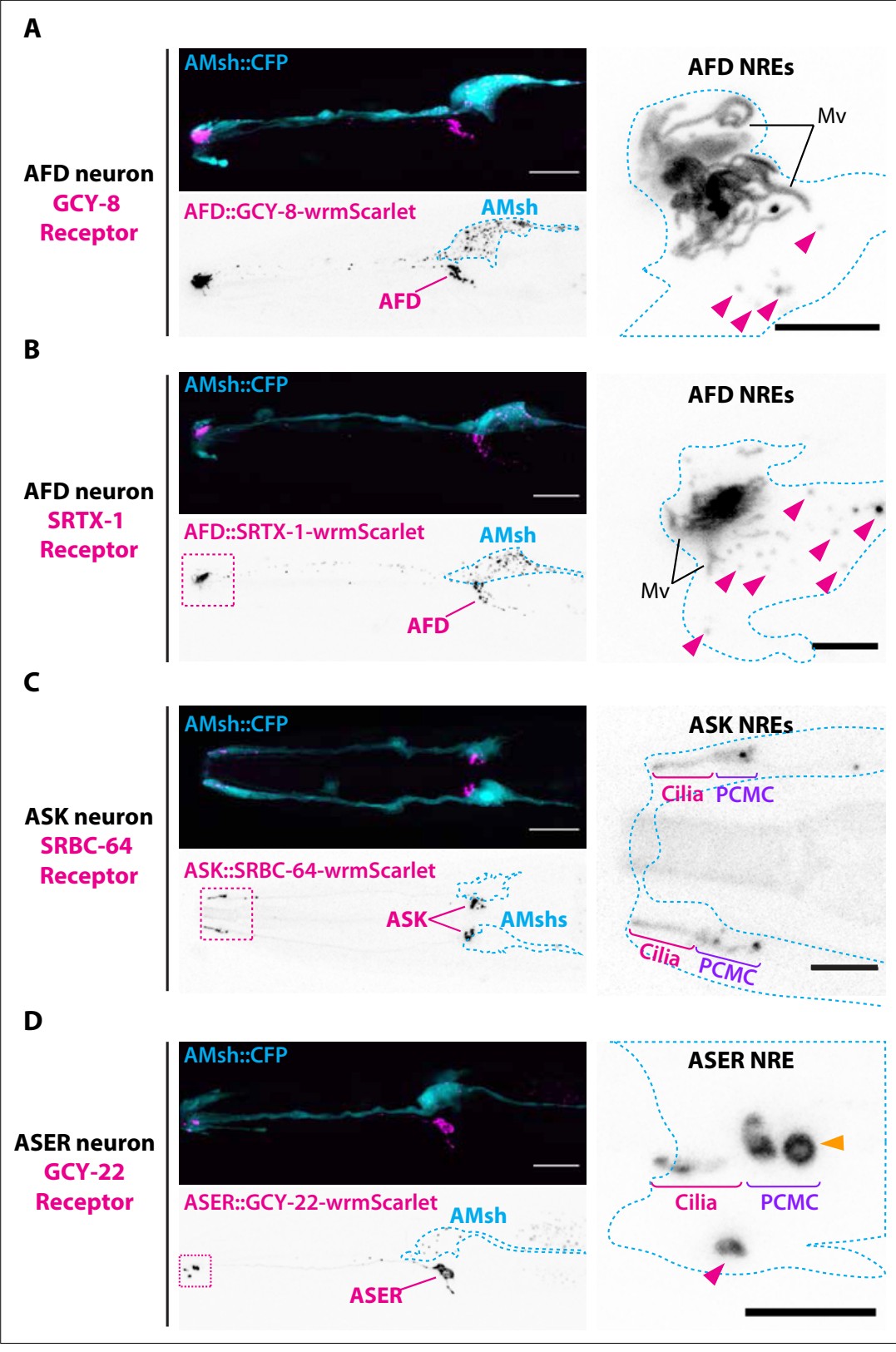

**Figure 3.** Endogenous ciliary membrane proteins are sorted to ectosomes and exported to AMsh. (**A**) GCY-8-wrmScarlet was cell-specifically expressed in AFD (driven by *gcy-8* promoter). GCY-8-wrmScarlet is enriched in AFD microvilli and AFD periciliary membrane compartment (PCMC). GCY-8-wrmScarlet-carrying extracellular vesicles (EVs) were also observed within AMsh cytoplasm, in the vicinity of the AFD nerve receptive endings and in AMsh

*Figure 3 continued on next page*

*Figure 3 continued*

cell body (magenta arrowheads). (**B**) SRTX-1-wrmScarlet was cell-specifically expressed in AFD (driven by *gcy-8* promoter). SRTX-1-wrmScarlet is enriched in AFD microvilli and PCMC, similarly to GCY-8. Within AMsh cytoplasm, SRTX-1-wrmScarlet-carrying EVs are observed in the vicinity of the AFD neuron receptive endings and in AMsh cell body (magenta arrowheads). (**C**) SRBC-64-wrmScarlet was cell-specifically expressed in ASK neurons (driven by *srbc-64* promoter). SRBC-64-wrmScarlet is observed in the ASK cilia proper (**C**) and PCMC but not in the cytoplasm of AMsh. (**D**) GCY-22-wrmScarlet was cell-specifically expressed in ASER (driven by *gcy-5* promoter). GCY-22-wrmScarlet is observed in ASER cilium tip and in ASER PCMC. ASER PCMC shows a rounded protrusion, which we consider as a recently excised EV (orange arrowhead). Within AMsh cytoplasm, a GCY-22-wrmScarlet-containing ectosome is located in the vicinity of the ASER PCMC. Few but large GCY-22-wrmScarlet-carrying EVs are observed in AMsh cell body. Scale bar: 20 μm in head images, 5 μm in insets.

but lost its enrichment at the cilia end (*Figure 4A*). In parallel, GCY-22-wrmScarlet export from ASER to AMsh was lost in *unc-101(m1)* (*Figure 4B*). Therefore, ectocytosis of GCY-22-wrmScarlet towards AMsh requires its sorting, trafficking, and accumulation in ASER cilia.

As the accumulation of GCY-22-wrmScarlet in ASER cilium might drive ectocytosis, we next used a strain where the *gcy-22* was tagged with GFP by CRISPR-Cas9 (*van der Burght et al., 2020*). Endogenously tagged GCY-22-GFP localized mostly to the cilia tip and to PCMC of ASER. Although not frequently, EVs containing GCY-22-GFP were observed in the cilia pore, as previously observed with GCY-22-wrmScarlet overexpression (*Figure 4C and F*). However, ectocytosis of GCY-22-GFP towards AMsh was not observed in wild-type animals (*Figure 4C and G*). Cilia trafficking mutants can cause pathological cargo accumulation at the cilia tip or in the PCMC according to mutant (*van der Burght et al., 2020*). In trafficking mutants, we asked whether GCY-22-GFP accumulation at the cilia tip or in the PCMC could induce GCY-22-GFP apical or basal ectocytosis, respectively. GCY-22 receptors rely on the OSM-3 kinesin for their entry and anterograde transport within ASER cilium (*van der Burght et al., 2020*). Ciliogenesis occurs in *osm-3(p802)* mutants, but the distal segment of the cilia is missing. Accordingly, we observed shorter cilia and enlarged PCMC in *osm-3(p802)* (*Figure 4D and E*). The accumulation of GCY-22-GFP at the cilia tip observed in controls was not observed in *osm-3(p802)* (*Figure 4B*). Instead, GCY-22-GFP accumulated at PCMC of *osm-3(p802)* mutants. As predicted, *osm-3(p802)* promoted ectocytosis of GCY-22-GFP towards AMsh (*Figure 4G*). CHE-3 is a dynein heavy chain involved in cilia retrograde traffic. *che-3(cas511)* mutants display severely truncated cilia (*Signor et al., 1999*; *Yi et al., 2017*; *Jensen et al., 2018*). In *che-3(cas511)* mutants, GCY-22-GFP accumulated in a strongly misshaped ASER cilium, including shorter cilia and enlarged PCMC (*Figure 4C, D and E*). Ectocytosis of GCY-22-GFP towards AMsh was increased, suggesting again that GCY-22-GFP accumulation at PCMC promotes its ectocytosis towards AMsh (*Figure 4G*). The BBSome subunit BBS-8 is dispensable for cilia assembly per se, but is required for cilia function by regulating protein trafficking, including receptor retrieval from cilia (*Blacque, 2004*; *Wei et al., 2012*). In IMCD3 cells, BBSome depletion reduced cilia retrieval of receptors but increased their ectocytosis from cilia tip (*Nager et al., 2017*). As predicted, *bbs-8(nx77)* increased GCY-22-GFP ectocytosis from ASER cilium towards the amphid pore (*Figure 4F*). GCY-22::GFP accumulation in ASER cilium was reduced in *bbs-8(nx77)*, suggesting that GCY-22::GFP escapes cilium by apical ectocytosis in the absence of BBSome retrieval, likely through ectocytosis (*Figure 4H, I and J*). Indeed, apical ectosomes were observed still attached

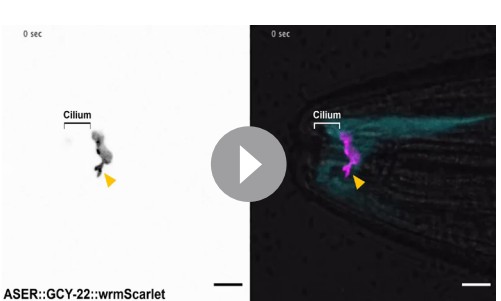

**Video 6.** Budding of basal ectosomes containing GCY-22-wrmScarlet originating from ASER cilium. An animal expressing GCY-22-wrmScarlet in ASER was immobilized with 10 mM tetramisole and recorded during 12 min. The acquisition plane corresponds to ASER periciliary membrane compartment (PCMC), for informative purposes ASER cilium location is indicated. Large basal ectosomes (~1 μm diameter) carrying GCY-22-wrmScarlet (magenta) are observed being released from the PCMC of ASER, released material ends up in AMsh cell (blue, AMsh::CFP cytoplasmic expression). GCY-22-wrmScarlet channel is shown in inverted LUT on the left. Scission events occur at t = 4 s and t = 348 s (orange arrow indicates the start of each scission events). Scale bar: 5 μm.
https://elifesciences.org/articles/67670/figures#video6

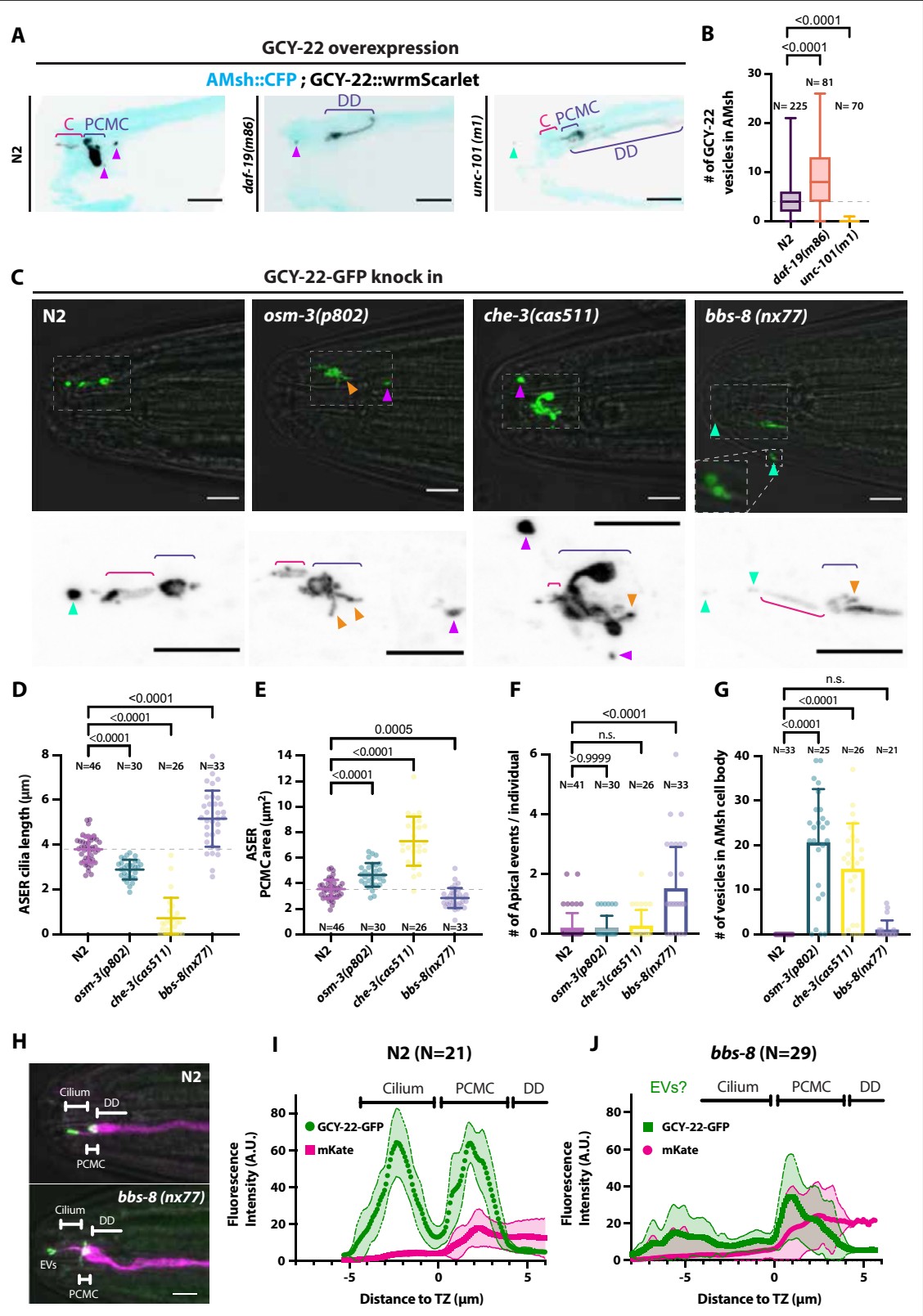

**Figure 4.** Ectocytosis of GCY-22-wrmScarlet is increased in *daf-19, osm-3, che-3, bbs-8* but reduced in *unc-101* mutants. (**A**) Representative images depicting overexpressed GCY-22-wrmScarlet distribution in ASER ciliary region for N2, *daf-19,* and *unc-101*. In N2, overexpressed GCY-22-wrmScarlet accumulated in the cilium (**C**) and in the periciliary membrane compartment (PCMC); however, it was absent from distal dendrite (DD). In *daf-19* mutants, ASER cilium was missing. GCY-22-wrmScarlet accumulated in an elongated ectopic compartment arising from ASER DD, an EV was observed in its

*Figure 4 continued on next page*

*Figure 4 continued*

vicinity within AMsh cytoplasm (magenta arrowhead). In *unc-101* mutants, GCY-22-wrmScarlet was observed weakly along ASER DD membrane, PCMC, and cilium with no enrichment in cilium. The ASER cilium was shorter. We observed one extracellular vesicle (EV) outside the animal (cyan arrowheads). Scale bar: 5 μm. (**B**) Number of large ectosomes containing GCY-22-wrmScarlet within AMsh cell. Box and whiskers plot represents their median number, the interquartile range, and the min/max values in N2, *daf-19*, and *unc-101*. The number of vesicles was increased in *daf-19* and decreased in *unc-101* mutants. Brown–Forsythe ANOVA, multiple comparisons corrected by Dunnett´s test. (**C**) In GCY-22-GFP knocked in strain in N2 background, GCY-22-GFP accumulated in the cilium and PCMC. In *osm-3*, GCY-22-GFP accumulated in ASER PCMC. PCMC shape was disrupted and displayed multiple protrusions (orange arrowheads). We observed more EVs in AMsh in the vicinity of ASER cilium (magenta arrowhead). In *che-3*, the ASER cilium proper was strongly shortened. GCY-22-GFP strongly accumulated in heavily disrupted PCMC displaying multiple protrusions filled with GCY-22-GFP. In *bbs-8* mutants, we observed more EVs in the amphid pore and outside (cyan arrowheads). ASER PCMC displayed abnormal shapes and often display protrusions. Scale bars: 5 μm. (**D**) The ASER cilium length was evaluated based on GCY-22-GFP staining of the cilium in 2D projections. Cilia length is shortened in *osm-3* and *che-3* and elongated and variable in *bbs-8* mutants. Brown–Forsythe ANOVA, multiple comparisons corrected by Dunnett´s test. (**E**) PCMC is increased in *osm-3* and *che-3* and reduced in *bbs-8* mutants. Brown–Forsythe ANOVA, multiple comparisons corrected by Dunnett´s test. (**F**) The number of apical EVs observed in each animals shows apical release occurs in N2, *osm-3*, and *che-3* but is potentiated in *bbs-8* mutant. Brown–Forsythe ANOVA, multiple comparisons corrected by Dunnett´s test. (**G**) The number of EVs observed in AMsh for each animal shows that basal release does not occur in N2. The number of EVs observed AMsh is increased in *osm-3* and *che-3* mutants. Kruskal–Wallis test, multiple comparisons corrected by Dunn´s test. (**H**) Fluorescence along the cilia was quantified in animals carrying GCY-22-GFP knock-in in N2 and *bbs-8* genetic background and an extrachromosomal for expression of mKate in ASER. Scale bar: 5 μm. (**I**) Linescans were traced for 21 N2 cilia and aligned on the transition zone based on drop in mKate signal. Average fluorescence standard deviation is plotted for mKate and GCY-22-GFP fluorescence intensities. It shows the accumulation of GCY-22-GFP fluorescence in PCMC and distal cilia. (**J**) Linescans were traced for 29 *bbs-8(nx77)* cilia and aligned on the transition zone based on drop in mKate signal. Average fluorescence standard deviation is plotted for mKate and GCY-22-GFP fluorescence intensities. It shows reduced GCY-22-GFP fluorescence along the cilia of *bbs-8(nx77)* and a highly variable distal cilia, representing elongated cilia, cilia with EVs attached to it, and EV detached from cilia.

or already separated from ASER (***Figure 4H***, bottom). Interestingly, we observed that ASER cilium displayed variable length potentially explained by this dynamic process of ectocytosis (***Figure 4D***). In vivo time-lapse acquisition of GCY-22-GFP in *bbs-8(nx77)* mutants showed ectosomes are released from the cilia tip at high rate (***Video 7***). Altogether, these results suggest that ectocytosis from ASER cilia contributes to reduce accumulation of GCY-22-GFP in cilia trafficking mutants.

Previous observations with GCY-22-GFP knock-in suggested that GCY-22-GFP ectocytosis from cilia tip occurs physiologically at low frequency but that ectocytosis towards AMsh only occurred secondarily to accumulations of GCY-22-GFP in ASER PCMC of *osm-3* and *che-3* mutants. To inves-

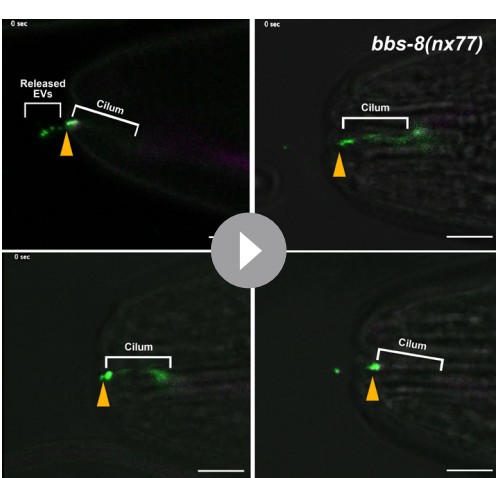

**Video 7.** Budding of apical ectosomes containing GCY-22-GFP originating from ASER cilia tip. Four examples of *bbs-8(nx77)* mutants expressing endogenously GCY-22-GFP. Each animal was immobilized with 10 mM tetramisole and recorded between 30 s and 4.5 min. Ectosomes shed continuously from ASER cilia tip (orange arrowheads) are sequentially released to the environment. Scale bar: 5 μm.
https://elifesciences.org/articles/67670/figures#video7

tigate whether other EV cargo or other neurons behave differently to GCY-22-GFP in ASER, we endogenously tagged *tsp-6* by CRISPR-Cas9 to fuse it with wrmScarlet. As predicted by single-cell mRNA sequencing data, TSP-6-wrmScarlet was expressed in several amphid neurons identified as ASK, ADF, ASJ, ASI, ASH, AWA, and AWC (***Figure 5A*** and ***Figure 5—figure supplement 1A and B***). EVs containing TSP-6-wrmScarlet were consistently observed in the amphid pore (1.42 ± 1.47 SD apical events/animal, N = 26). Video capture shows release of TSP-6-wrmScarlet EVs occurs in bolus events in a time frame of seconds (***Video 8***). EVs carrying TSP-6-wrmScarlet were also observed within AMsh cytoplasm, suggesting that ectocytosis of TSP-6-wrmScarlet towards AMsh occurs physiologically from TSP-6-expressing neurons (***Figure 5B***). All other observations made with GCY-22-GFP in cilia trafficking mutants were replicated with TSP-6-wrmScarlet animals: ectocytosis towards the amphid pore was increased approximately three times in *bbs-8(nx77)* and unchanged in *osm-3(p802)* and *che-3(cas511)*. Ectocytosis towards AMsh was increased approximately four times in

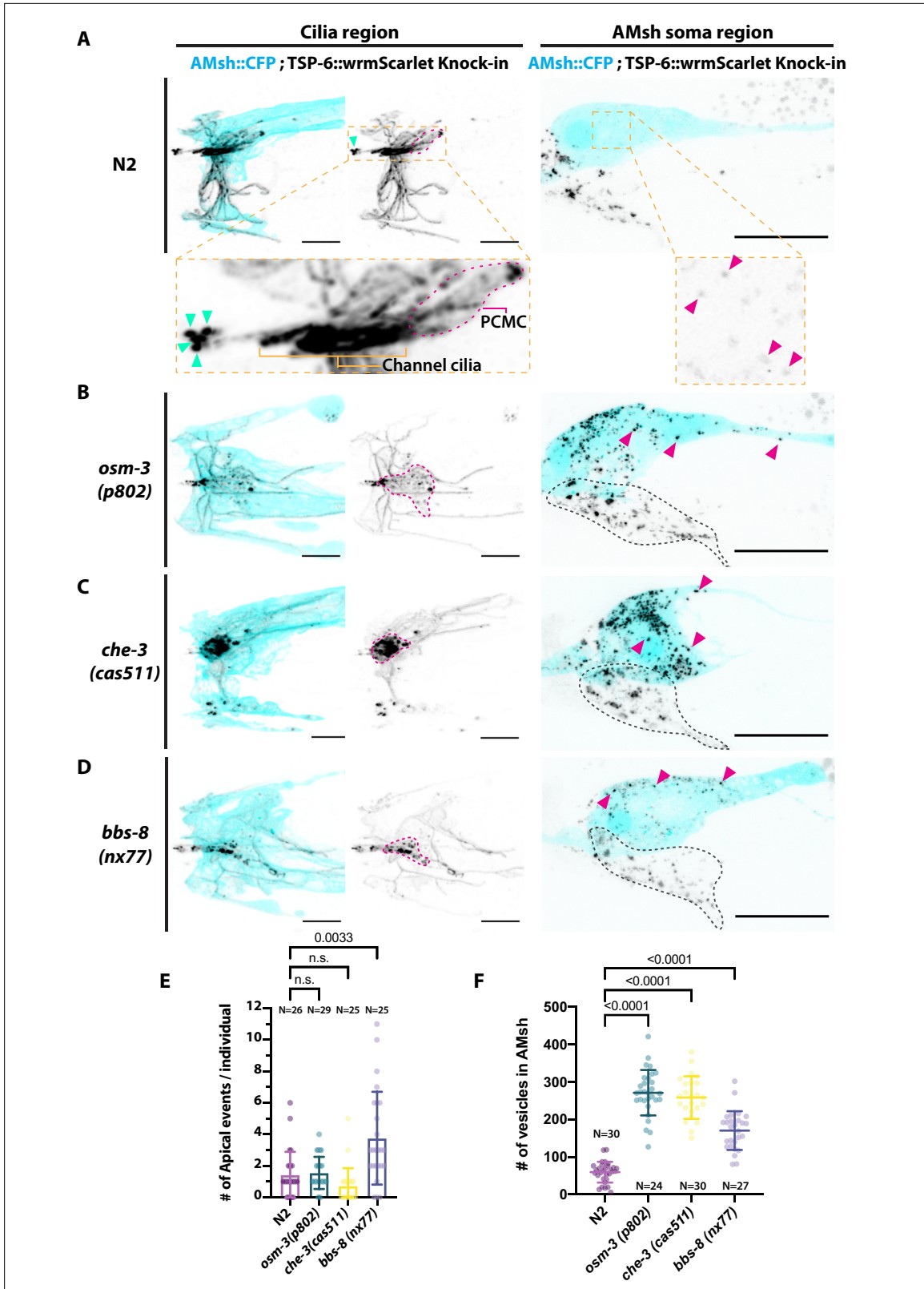

**Figure 5.** Ectocytosis of TSP-6-wrmScarlet to AMsh is increased in *osm-3, che-3, and bbs-8* while ectocytosis of TSP-6-wrmScarlet to the amphid pore and outside is increased in *bbs-8*. (**A**) In TSP-6-wrmScarlet knocked in strain in N2 background, TSP-6-wrmScarlet accumulated in the cilium and periciliary membrane compartment (PCMC) of the channel amphid neurons (Inset) as well as in AWC and AWA cilia. Several extracellular vesicles (EVs) were observed in amphid channel (cyan arrowheads). Left panel, scale bar: 5 µm. TSP-6-wrmScarlet was also observed in EVs located within the AMsh

*Figure 5 continued on next page*

*Figure 5 continued*

cytoplasm (magenta arrowheads). Right panel, scale bar: 20 µm. (**B**) In *osm-3*, TSP-6-wrmScarlet accumulated in PCMCs area (delimited by dashed magenta line). We observed more EVs in AMsh cell body (magenta arrowhead). The location of amphid neurons cell bodies is delimited by a black dashed line. (**C**) In *che-3*, TSP-6-wrmScarlet strongly accumulated in PCMCs. We observed more EVs in AMsh cell body (magenta arrowhead). (**D**) In *bbs-8* mutants, we observed more EVs in the amphid pore (cyan arrowheads) and in the AMsh cell body (magenta arrowheads). (**E**) The number of apical EVs observed in each animals shows that apical release occurs in N2, *osm-3*, and *che-3*. Apical release is potentiated in *bbs-8* mutants. Brown–Forsythe ANOVA, multiple comparisons corrected by Dunnett´s test. (**F**) The number of EVs observed in AMsh for each animals shows that TSP-6-wrmScarlet export to AMsh occurs in N2. This number of EVs observed AMsh is increased in *osm-3*, *che-3,* and *bbs-8* mutants. Brown–Forsythe ANOVA, multiple comparisons corrected by Dunnett´s test.

The online version of this article includes the following figure supplement(s) for figure 5:

**Figure supplement 1.** (**A**) Expression pattern of TSP-6-wrmScarlet in gene-edited strain and export of TSP-6-wrmScarlet to AMsh.

**Figure supplement 2.** Periciliary membrane compartment (PCMC) and ectosome size are influenced by expression of TSP-6 and GCY-22.

*osm-3(p802)* and *che-3(cas511)* (*Figure 5C and D*). In addition, we observed ectocytosis of TSP-6-wrmScarlet towards AMsh increased ~2.5 times in *bbs-8(nx77)* mutants; something also observed for GCY-22-GFP but not reaching significance. Although quantification is tricky, the channel cilia of TSP-6-expressing neurons appeared slightly shorter in *osm-3(p802)* and severely truncated in *che-3(cas511)*, as previously described. We observed abnormal accumulation of TSP-6-wrmScarlet in area where PCMCs are located in *osm-3(p802)* and *che-3(cas511)* (*Figure 5A*). We also observed extended AWA branches in *bbs-8(nx77)* and *che-3(cas511)* but not in *osm-3(p802)*. Altogether, the TSP-6-wrmScarlet knock-in strain confirms that local cargo ectocytosis contributes to reduce pathological accumulation of TSP-6-wrmScarlet in cilia trafficking mutants.

## Cargo composition can alter EV budding dynamics

PCMC expansion was observed in GCY-22-wrmScarlet overexpression strains but not in GCY-22-GFP knock-in strain. Interestingly, PCMC expansions were also observed in GCY-22-GFP knock-in strains in the *che-3(cas511)* and *osm-3(p802)* mutants (*Figure 4C*). We compared these PCMC expansion (*Figure 5—figure supplement 2A*). Overexpression of GCY-22 in ASER PCMC can lead to similar PCMC expansion than pathological accumulation of knocked in GCY-22-GFP in ASER PCMC induced by *che-3*. Therefore, cargo accumulation in PCMC observed in cilia trafficking mutants can also occur in overexpression strains potentially because of a saturation of the ciliary trafficking machinery.

In contrast, overexpression of TSP-6-wrmScarlet in ASER did not induce such PCMC expansion (*Figure 5—figure supplement 2A*). What could explain this difference between cargo overexpression? We observed that overexpression of GCY-22-wrmScarlet in ASER produced few but large EVs in AMsh while overexpression of TSP-6-wrmScarlet in ASER produced small but numerous EVs in AMsh (*Figures 3D and 2C*). Build-up of GCY-22-wrmScarlet in PCMC because of its overexpression was observed to slowly (~ 1 event/14.66 min, N = 3) bud into large (~1 µm) ectosomes (*Video 6*, *Figure 3D*, and *Figure 5—figure supplement 2B*), while build-up of TSP-6-wrmScarlet in PCMC because of its overexpression bud frequently (~ 1 event/1 min, N = 3) into ( <500 nm) ectosomes (*Video 5*, *Figure 2C*, and *Figure 5—figure supplement 2B*). Ectosome size difference was maintained in knocked in strains (*Figure 2C* and *Figure 5—figure supplement 2B*). Altogether, these observations suggested us that GCY-22 and/or TSP-6 accumulation in ASER cilium could alter ectosome budding dynamics and ultimately EVs size. To explore this possibility, we co-expressed GCY-22-GFP together with TSP-6-wrmScarlet in ASER. We observed a poor overlap of the two membrane proteins within ASER cilium (*Figure 5—figure supplement 2C*). Both markers

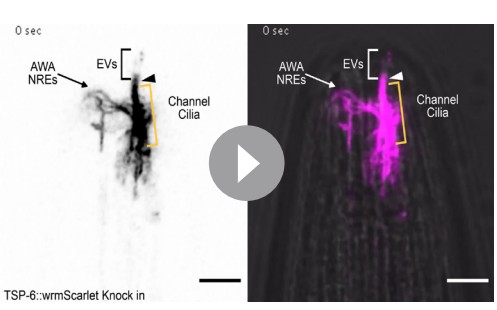

**Video 8.** Budding of apical ectosomes containing TSP-6-wrmScarlet originating from one or several cilia of ASK, ADF, ASJ, ASI, and ASH. An animal expressing endogenous TSP-6 levels (tsp-6:wrmScarlet knock-in strain) was immobilized with 10 mM tetramisole and recorded during approximately 2,5 min. Ectosomes shed continuously from the cilia tip (white arrowheads) and are trapped within the pore channel occasionally being released to the environment. Scale bar: 5 µm.
https://elifesciences.org/articles/67670/figures#video8

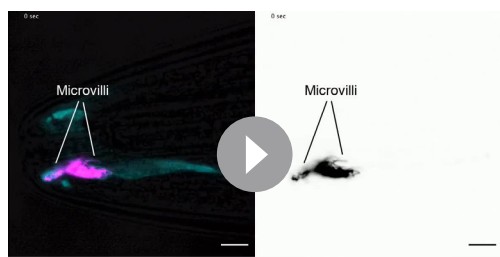

pGCY-8::TSP-6::wrmScarlet

**Video 9.** AFD microvilli fragments are released and subsequently captured by AMsh glia. An animal expressing TSP-6-wrmScarlet in AFD was immobilized with 10 mM tetramisole and recorded during approximately 8 min. Microvilli tips sometimes get detached from AFD nerve receptive endings (NREs) and are subsequently captured by AMsh (blue, AMsh::CFP cytoplasmic expression). Scission events occur at t = 125 s (yellow arrowhead indicates the start of the scission event). Scale bar: 5 μm.

https://elifesciences.org/articles/67670/figures#video9

were sorted to EVs and exported by ASER into AMsh, 75 % of these EVs observed in the surroundings of ASER cilium were carrying TSP-6-wrmScarlet alone, 23 % of them were carrying TSP-6-wrmScarlet together with GCY-22-mEGFP and very few vesicles were observed with only GCY-22-mEGFP. Interestingly, the EVs carrying GCY-22-mEGFP and TSP-6-wrmScarlet together have an intermediate size between the ~1 μm of ectosomes carrying GCY-22-wrmScarlet alone and the ~400 nm ectosomes carrying TSP-6-wrmScarlet alone (*Figure 5—figure supplement 2D*). These observations suggest that accumulation of some cargo affects budding dynamics and that cargo interactions might set the final EV diameter.

## Pruning by AMsh glia is not necessary for the production of EVs by AFD

The previous observations suggested that budding dynamics mostly depend on cilia cargo. However, budding from PCMC is coupled to AMsh phagocytosis. What is the contribution of AMsh glia in budding? In several animal models, glial cells actively prune excess synapses, axonal projections, dendritic spines, and sensory endings of neurons (*Wilton et al., 2019*). Using TSP-7-wrmScarlet export from AFD, we explored the role of AMsh in EV capture. Time-lapse recordings from animals expressing TSP-6-wrmScarlet in AFD neurons showed that many of the TSP-6-wrmScarlet EVs seemed to originate from AFD microvilli endings. Additionally, we could observe elongation and retraction of microvilli (*Video 9*). This pruning of AFD microvilli was recently observed by *Raiders et al., 2021*. We first asked whether pruning of AFD microvilli was strictly necessary to observe export of TSP-7-wrmScarlet from AFD to EVs in AMsh. To this end, we used a mutant for TTX-1, a transcription factor required for correct morphogenesis of AFD terminals. *ttx-1(p767)* mutants are defective for AFD microvilli formation but maintain their cilia (*Perkins et al., 1986*; *Satterlee et al., 2001*). In *ttx-1(p767)* mutants, TSP-7-wrmScarlet remained enriched in AFD receptive endings lacking microvilli (*Figure 6A*). TSP-7-wrmScarlet was still exported to AMsh in the *ttx-1(p767)*, although the number of TSP-7-wrmScarlet EV and their fluorescence was strongly reduced (*Figure 6B* and *Figure 6—figure supplement 1A and B*). Therefore, the AFD microvilli are important but not absolutely required for TSP-7-wrmScarlet export from AFD to EVs in AMsh. We next asked whether the export of TSP-7-wrmScarlet from AFD to EVs in AMsh required the full amphid sensilla structure or only the close apposition of AFD to AMsh. To answer this question, we used the *dyf-7(m537)* mutants where the AMsh still ensheathes AFD NREs but in an ectopic location inside the head (*Heiman and Shaham, 2009*). Despite this displacement of AFD NREs in *dyf-7* mutants, the transfer of TSP-7-wrmScarlet from AFD to EVs in AMsh still occurred: we observed the typical fluorescent EVs throughout the cell body of AMsh (*Figure 6C*). Therefore, export of TSP-7-wrmScarlet only requires the AFD receptive endings to contact AMsh, independently of a proper microvilli morphology and location of the amphid sensilla.

We next asked whether the presence of AMsh is required for AFD to produce TSP-7-wrmScarlet EVs. For this, AMsh was genetically ablated by means of diphtheria toxin (DTA) expression. In the strain we used (OS2248), amphid neurons and their receptive endings properly develop before AMsh ablation occurs in late embryogenesis (*Bacaj et al., 2008*). In the absence of AMsh, TSP-7-wrmScarlet export still occurred from AFD receptive endings to EVs located in other neighboring cells, presumably hypoderm cells from the head (*Figure 6D*, right panels). Therefore, pruning of AFD microvilli by AMsh or an instructive signal from AMsh is not strictly necessary for AFD to produce TSP-7-wrmScarlet EVs. Instead, the export of TSP-7-wrmScarlet simply requires the EVs produced by AFD to be captured by any competent neighboring cell type. Among the local cells undergoing constitutive endocytic

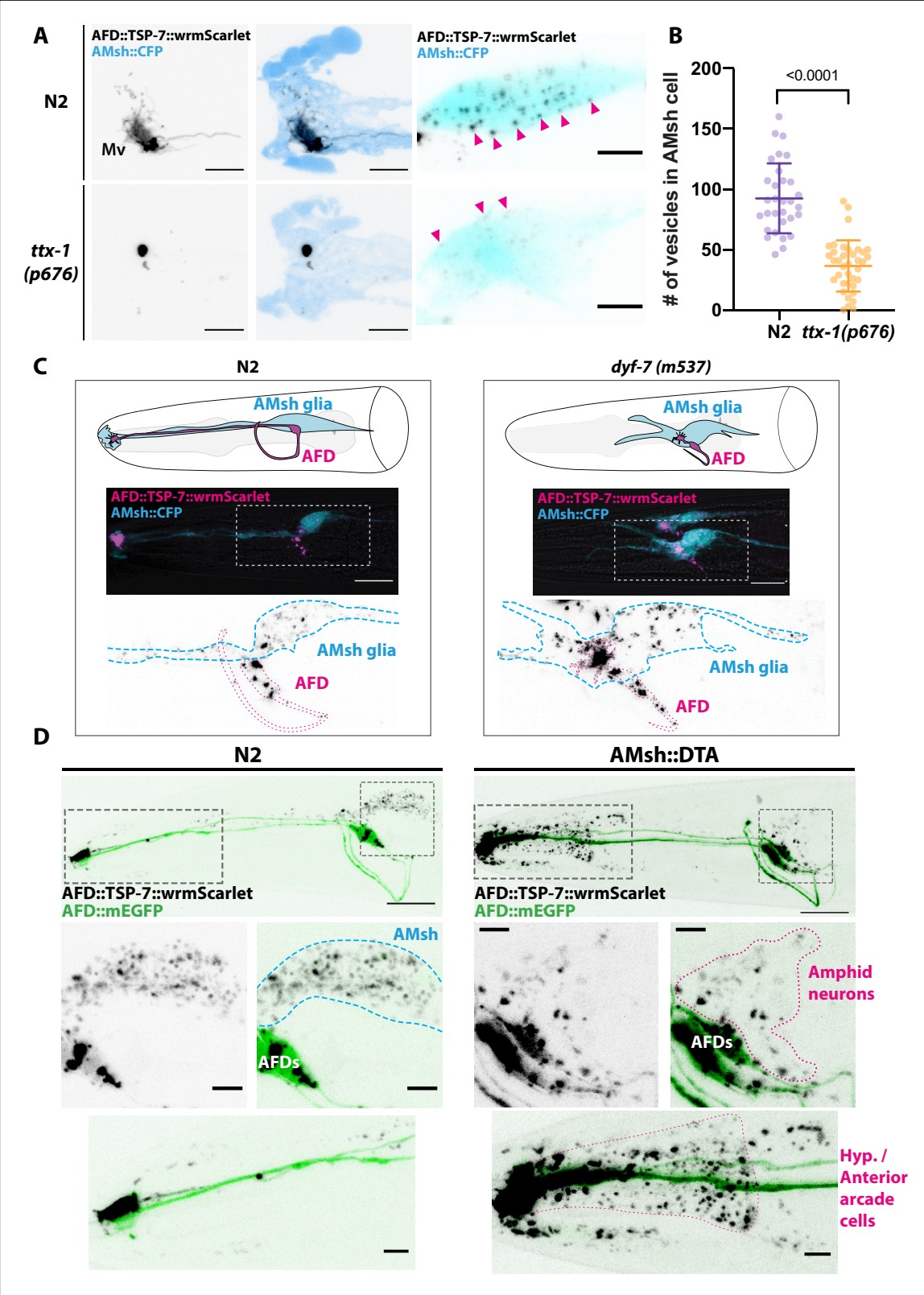

**Figure 6.** Position and presence of glia are not necessary for extracellular vesicle (EV) production and export to occur. (**A**) In *ttx-1(p767)* mutants, the microvilli (Mv) disappeared but TSP-7-wrmScarlet remained enriched in the remaining AFD distal region. Scale bar: 5 µm. (**B**) Export of TSP-7-wrmScarlet from AFD to AMsh was quantified by counting the number of EVs exported to AMsh. TSP-7-wrmScarlet export is decreased in *ttx-1(p767)* mutants. Unpaired t-test. (**C**) In *dyf-7(m537)* mutants, TSP-7-wrmScarlet remained enriched in AFD receptive endings although the receptive ending was displaced

*Figure 6 continued on next page*

Figure 6 continued

posteriorly in the animal's head but still embedded within AMsh. TSP-7-wrmScarlet was still exported to AMsh cell body in a similar manner to wild-type controls. Scale bar: 20 μm. (D) Representative images displaying differential tissue capture of EVs when glia is ablated genetically post-embryogenesis. Animals expressed AFD::mEGFP and AFD::TSP-7-wrmScarlet. TSP-7-wrmScarlet is enriched in AFD receptive end in both experimental conditions. EVs are exported to AMsh in control conditions. In the absence of glia, EVs containingTSP-7-wrmScarlet were still produced but were exported to large cells at the surface of the nose, likely the hypodermal cells. TSP-7-wrmScarlet was also exported to amphid sensory neurons. Scale bar: 20 μm for top head images, 5 μm for insets.

The online version of this article includes the following figure supplement(s) for figure 6:

**Figure supplement 1.** In addition to counting extracellular vesicles (EVs) in AMsh, we quantified export of TSP-7-wrmScarlet from AFD to AMsh by extracting the fluorescence of all EVs in AMsh.

**Figure supplement 2.** Amphid sheath ablation reroutes intake of extracellular vesicles (EVs) to nearby amphid neurons.

activity and located in the proximity of AFD NREs are all the ciliated amphid neurons. Interestingly, in the absence of AMsh, a fraction of TSP-7-wrmScarlet is exported from AFD to EVs located in the cytoplasm of a subset of ciliated amphid neurons also expressing the osm-3p::mEGFP transgene (*Figure 6—figure supplement 2*). Therefore, amphid neurons can also take up TSP-7-wrmScarlet at their receptive endings in the absence of AMsh. Altogether, these observations suggest that EVs produced intrinsically by AFD are physiologically taken up by the scavenging activity of AMsh as long as AFD receptive ends are embedded within AMsh. In the absence of AMsh, any local endocytic activity can take up these EVs.

## Phagocytic activity in AMsh is required to maintain sensory cilia shape and function

Our previous results highlight the active role of neurons for basal ectosome production. However, ectosomes do not accumulate in the amphid sensilla of hermaphrodites, highlighting their efficient clearance by AMsh. This clearance could occur passively by phagocytosis of passing-by EVs; in this case, an AMsh endocytic defect would lead to EV accumulation in the lumen of the amphid sensilla, between amphid cilia and AMsh. Alternatively, as PCMC and AMsh plasma membranes are closely associated, PCMC ectocytosis might occur simultaneously to glial phagocytosis. Such mechanism would facilitate ectosome clearance and limit the risk of ectosomes accumulating between cilia and glia. In this case, an AMsh endocytic defect would lead to abnormal cilia shape caused by unsolved ectosome uptake. To test these two models, we reduced AMsh endocytic activity by overexpressing cell-specifically a dominant negative construct for DYN-1. In human macrophages, Dynamin 2 is recruited early during phagosome formation and contributes to phagosome scission from plasma membrane (*Marie-Anais et al., 2016*). Dynamin 2 dominant negative mutation (K44A) abolishes GTP binding activity and disturb pseudopod extension (*Gold et al., 1999*). Similar dominant mutations in its *C. elegans* ortholog *dyn-1(G40E and K46A)* disturb engulfment rate, RAB-5 recruitment, phagosome maturation, and degradation (*Kinchen et al., 2008*; *He et al., 2010*). As expected, if ASER PCMC ectocytosis was coordinated with AMsh phagocytosis, we observed abnormal filopodia-like protrusions still attached to the ASER PCMC when DYN-1(K46A) was overexpressed in AMsh (*Figure 7A*). We never observed EVs accumulating between ASER and AMsh. Instead, many tiny vesicles were observed in the vicinity of ASER cilium, sometimes generating string of pearls within AMsh cytoplasm (*Figure 7—figure supplement 1A'*). To assess this non-cell-autonomous effect of AMsh::DYN-1(K46A) on other amphid neurons, we examined the morphology of ASH, AFD, and AWC NREs. In agreement with ASER results, we saw expression of DYN-1(K46A) in AMsh induced abnormal filopodia-like protrusions still attached to ASH PCMC (*Figure 7B*). Expression of DYN-1(K46A) in AMsh induced abnormal AWC cilia where the wing-like sheets are replaced by branches; filopodia-like protrusions still attached to AWC PCMC were observed in 11 of the 14 AWC cilia imaged (*Figure 7C*). Expression of DYN-1(K46A) in AMsh induced abnormal AFD microvilli architecture as well as abnormal filopodia-like protrusions attached to AFD PCMC (*Figure 7D*). We never observed EVs accumulating between ASH, AFD, AWC, AFD NREs, and AMsh, suggesting that the EVs produced are not accumulating outside the glia. Instead, we observed filopodia-like protrusion remaining attached to the PCMC. Expression of DYN-1(K46A) disturbs a whole set of AMsh functions, including phagocytosis. Therefore, filopodia-like protrusion originating from PCMC might correspond to ectosomes not readily pinched. We observed that DYN-1(K46A) affected AMsh position and shape, suggesting that it modifies AMsh cell biology. Previously,

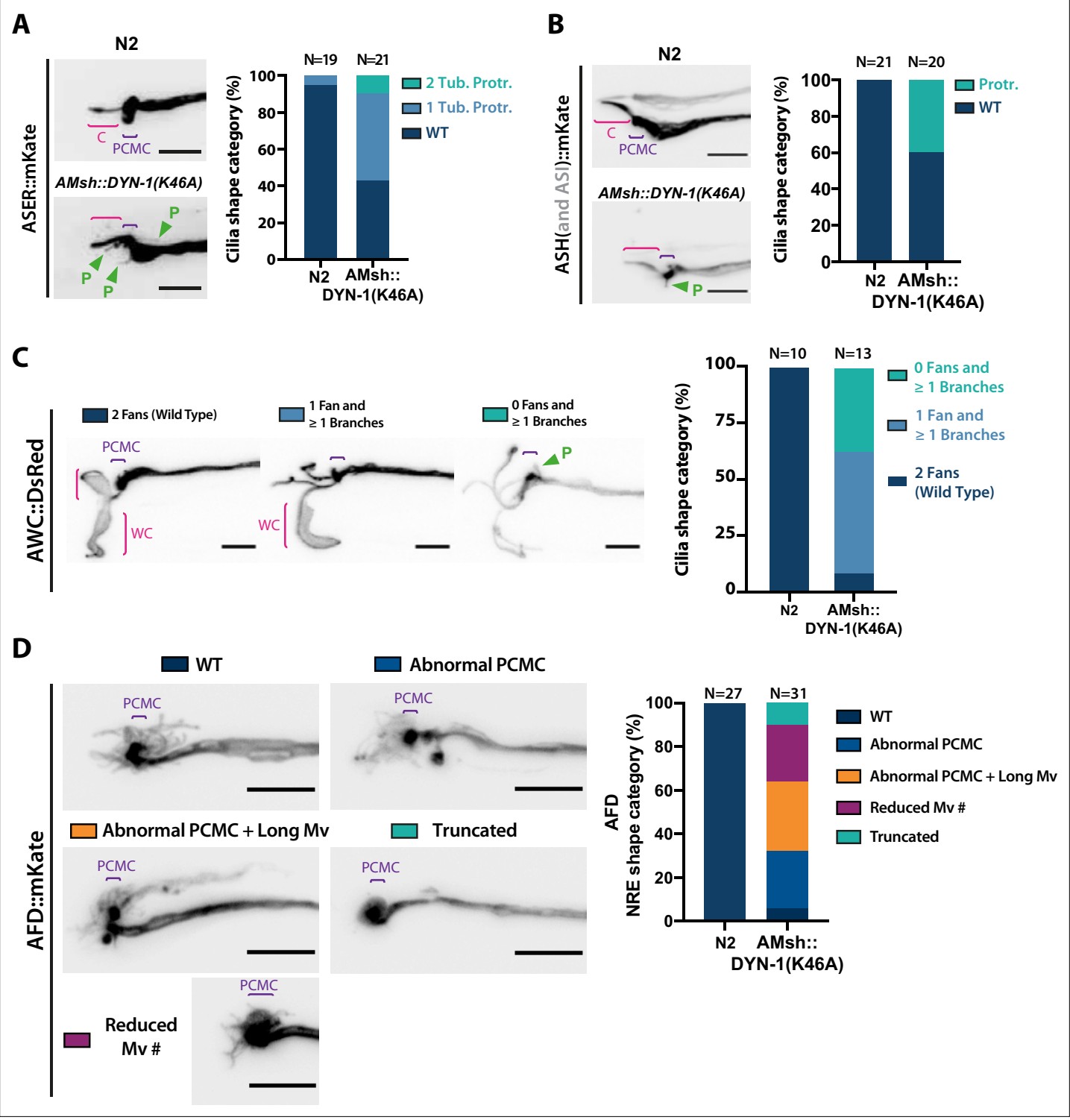

**Figure 7.** AMsh glia phagocytic activity is required to maintain a proper sensory cilia structure. (**A**) We examined ASER cilia shape in N2 and in AMsh::DYN-1(K46A) transgenics animals expressing mKate in ASER. Three categories were made according to ASER cilia shape: animals displaying zero (WT), one, or two filopodia-like protrusions still connected to periciliary membrane compartment (PCMC) (C: cilium; P: filopodial-like protrusion). The percentage of each cilium shape category is given for each genotype. Expression of DYN-1(K46A) transgene in AMsh strongly increased the number of animals showing PCMC protrusions. (**B**) We examined ASH cilia shape in N2 and in AMsh::DYN-1(K46A) transgenic animals expressing mKate in ASH. Two categories were made according to ASH cilia shape: animals displaying, or not, filopodia-like protrusions still connected to PCMC. AMsh::DYN-1(K46A) strongly increased the number of animals showing PCMC protrusions. (**C**) We examined AWC wing cilia (WC) with their characteristic

*Figure 7 continued on next page*

*Figure 7 continued*

membranous expansions in N2 and in AMsh::DYN-1(K46A) transgenics animals expressing DsRed in AWC. Three categories were made according to AWC cilia shape: animals displaying two WC (as wild type), animals displaying one WC and branches instead of the second WC, and animals displaying only branches instead of two WC. In addition, filopodia-like protrusions still connected to PCMC were observed in 11 of 14 AWC animals expressing AMsh::DYN-1(K46A). AMsh::DYN-1(K46A) transgene strongly increased the number of animals showing abnormal AWC cilia; however, AWC cilia was always maintained. (**D**) Nerve receptive ending (NRE) shape of AFD in N2 and in AMsh::DYN-1(K46A) transgenics expressing mKate in AFD. Five categories were established: wild-type phenotype (WT), abnormal AFD PCMC, abnormal PCMC + elongated microvilli (PCMC + Long Mv), reduced number of microvilli (Less Mv#), and full loss of microvilli (Truncated). AMsh::DYN-1(K46A) transgene strongly increased the number of animals showing abnormal AFD cilia; however, AFD NREs are maintained in 90% of the animals. Scale bar: 5 µm.

The online version of this article includes the following figure supplement(s) for figure 7:

**Figure supplement 1.** ASER sensory cilia structure is affected when AMsh phagocytic activity is disturbed.

it was shown that ablation of AMsh, cell-specific expression of RAB-1(S25N) causing exocytosis defect or a *pros-1* mutation causing secretome defect all induced severe truncation of AWC and AFD NREs (*Bacaj et al., 2008*; *Singhvi et al., 2016*; *Wallace et al., 2016*). We observed severe microvilli truncation in ~10% of AFD NRE, and we did not observe truncations of AWC wing cilia.

## Glial phagocytosis plays a role in sensory function

We reasoned that the aforementioned abnormal cilia structure in AMsh::DYN-1(K46A)-expressing animals could lead to defects in the sensory perception of the amphid sensory neurons. We explored this possibility by using well-described chemotactic assays for the chemosensory neurons ASER, ASH, AWC, and thermotactic assay for the temperature-sensing AFD neurons (*Bargmann, 2006*); for more information, see Materials and methods section and *Figure 8—figure supplement 1*. *C. elegans* shows a preference for NaCl concentrations associated to their cultivation on food, a behavior that depends on ASE sensory neurons. Within a linear NaCl gradient, we observed a loss of attraction to 50 mM salt in animals expressing DYN-1(K46A) in AMsh (*Figure 8A*). When placed into a gradient of the aversive copper ions ($Cu^{2+}$), *C. elegans* navigates towards low concentrations, a behavior that depends on the ASH nociceptor neurons. Within a linear $Cu^{2+}$ gradient, animals expressing DYN-1(K46A) in AMsh completely lost repulsion to $Cu^{2+}$ (*Figure 8B*). When placed into a gradient of the attractive odorant isoamyl alcohol (IAA) *C. elegans* navigates toward the IAA source, a behavior that depends on AWC odorant neurons. Animals expressing DYN-1(K46A) in AMsh maintained a normal attraction to IAA (*Figure 8C*). When placed into a temperature gradient, *C. elegans* shows a preference for temperatures associated to their cultivation on food, a behavior that depends on AFD sensory neurons. Within a temperature gradient, animals expressing DYN-1(K46A) in AMsh maintained normal thermotaxis to their cultivation temperature (*Figure 8D*). These results imply that ASER and ASH sensory function are altered non-cell autonomously by expression of DYN-1(K46A) in AMsh, while AFD and AWC response were unaffected. Again, these results contrast with the effects on sensory response in the absence of AMsh or in defective AMsh exocytosis or secretome (*Bacaj et al., 2008*; *Singhvi et al., 2016*; *Wallace et al., 2016*). Downstream to cilia response to cues, sensory neurons depolarize and signal to a neuronal circuit that mediate behavioral response. To confirm signaling downstream to cilia response was functional in ASH, we used channelrhodopsin (ChR2). In this way, we can stimulate ASH independently of any sensory cues and sensory machinery. We did not observe any abnormal response to blue light exposure in animals expressing DYN-1(K46A) in AMsh (*Figure 8E*). Therefore, disturbing glial phagocytic function affects cilia shape and sensory perception, but downstream signal transduction, including sensory neuron depolarization, neurotransmission, and signal integration by the nervous system, remains unaltered.

## Discussion

Several early arguments suggested that EVs produced by *C. elegans* males correspond to ectosomes shed from the cilia of the male neurons (*Wang et al., 2014*). Simultaneous to what we describe, new observations suggested that ciliary ectosomes may be shed from two sites of the cilia of CEM neurons (*Wang et al., 2021*). Our results confirm and extend these conclusions: many – potentially all – hermaphrodite ciliated neurons can produce ciliary ectosomes from their cilia when ciliary membrane proteins are overexpressed. Using in vivo recordings and volumetric confocal acquisitions, we could

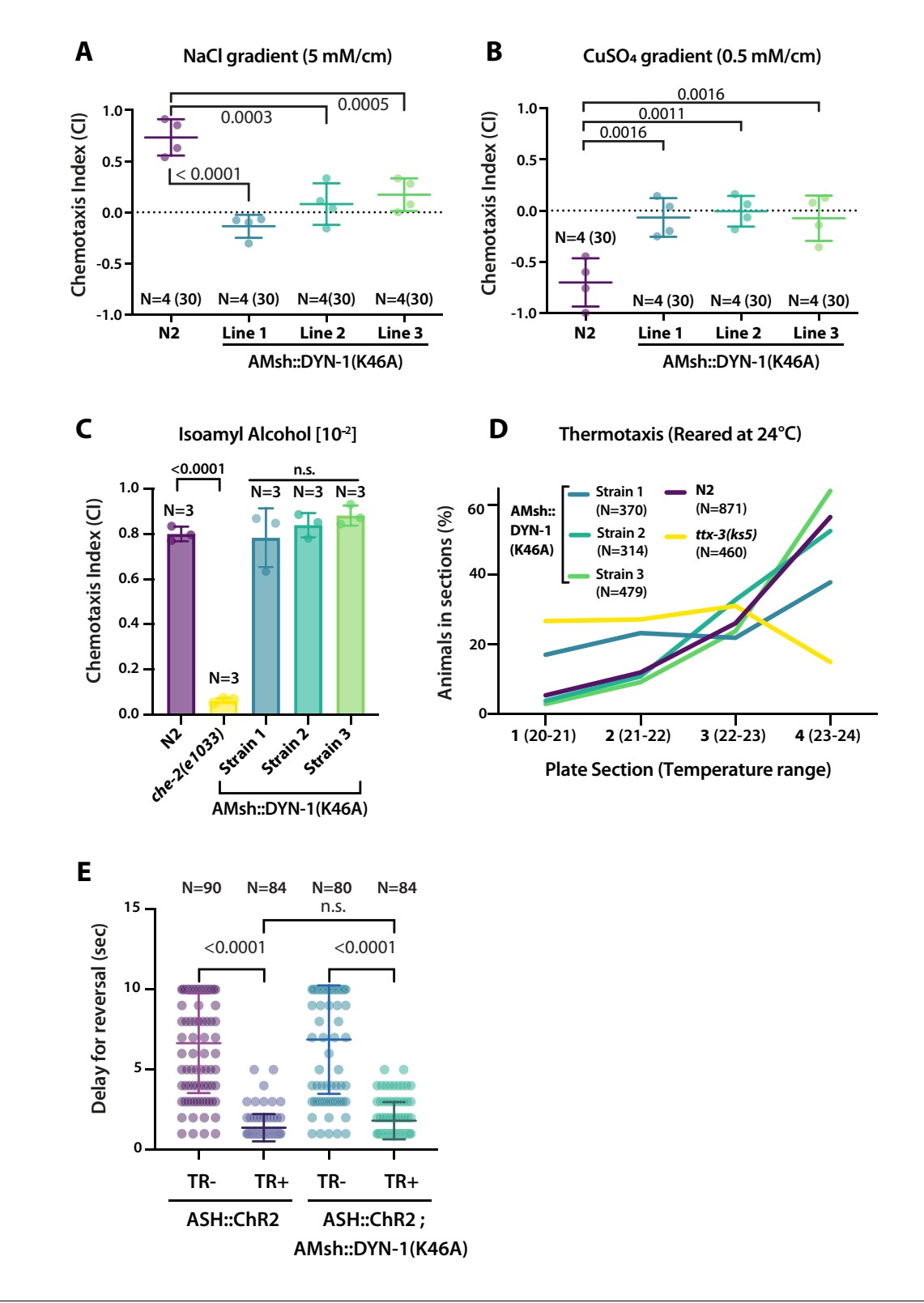

**Figure 8.** AMsh phagocytic activity affects ASER and ASH sensory functions. (**A**) Chemotactic indexes of four independent assays made with 30 N2 or AMsh::DYN-1(K46A) transgenics in linear 5 mM/cm NaCl gradients. The chemoattraction to NaCl was lost in transgenics expressing AMsh::DYN-1(K46A). One-way ANOVA, multiple comparisons corrected by Tukey test. (**B**) Chemotactic indexes of four independent assays made with 30 N2 or AMsh::DYN-1(K46A) transgenics in linear 5 mM/cm CuSO$_4$ gradients. Animals expressing AMsh::DYN-1(K46A) did not show avoidance behavior to CuSO$_4$. One-way

*Figure 8 continued on next page*

*Figure 8 continued*

ANOVA, multiple comparisons corrected by Tukey test. (**C**) Chemotactic indexes of three independent assays made with an average of >100 N2, *che-2*, or AMsh::DYN-1(K46A) transgenics in a gradient of the volatile attractant isoamyl alcohol (IAA). The top of the gradient was spotted with [10⁻²] IAA. AMsh::DYN-1(K46A) did not affect IAA chemotaxis. One-way ANOVA, multiple comparisons corrected by Dunnett's test. (**D**) Thermotactic behavior in one temperature gradient assays made with >400 N2, *ttx-3(ks5)* or AMsh::DYN-1(K46A) transgenics. AMsh::DYN-1(K46A) did not consistently affect thermotaxis. (**E**) Transgenic animals expressing ChR2(H134R) in ASH sensory neurons (ASH::ChR2(H134R); *lite-1*) exhibit fast reversal (minimum 1–2 backward head swings) in response to blue light exposure (15 mw/mm²). This response was only observed when animals were raised in the presence of trans-retinal (TR+). Control groups were done using the same strain and same stimulation but were raised in the absence of trans-retinal (-TR). Expression of AMsh::DYN-1(K46A) does not modify this avoidance response. Kruskal–Wallis test, multiple comparison corrected by Dunn's test.

The online version of this article includes the following figure supplement(s) for figure 8:

**Figure supplement 1.** Experimental procedure followed for chemotaxis and thermotaxis assays.

identify the location for the biogenesis and excision of ciliary ectosomes. The fate of these ectosomes varies according to the location of their biogenesis: basal ectosomes budding from the PCMC is simultaneously captured by the contacting glial cells, while apical ectosomes shed from the cilia tip were environmentally released, exiting through the pore of the sensilla (*Figure 9*). Basal ectosomes phagocytosed by glial cells are likely directed to the endolysosomal pathway.

Does ciliary ectocytosis occur in physiological conditions? Physiological production of ciliary ectosome by neurons is documented in human photoreceptor (*Salinas et al., 2017*). Using knock-in strains, we show that ectosomes carrying TSP-6-wrmScarlet or GCY-22-GFP bud from cilia tip in the absence of overexpression artifacts. These observations suggest physiological production of EVs by several, if not all, ciliated neurons of hermaphrodite and male. We demonstrate that lipophilic DiI is captured by ciliated amphid neurons and exported from them to AMsh. Importantly, this assay shows that ciliary membrane is exported to AMsh in physiological conditions, arguing for a physiological export of ciliary material by ectocytosis to the supporting glia. In addition, we observe filopodia-like protrusion originating from PCMC in AMsh::DYN-1(DN), suggesting that basal ectocytosis occurs in the absence of pathological cargo accumulation in PCMC. However, overexpression of male ciliary cargoes leads to massive apical ectocytosis by *C. elegans* male ciliated neurons without observed basal release to the supporting glia (*Wang et al., 2014*). In physiological conditions, EVs carrying TSP-6-wrmScarlet or GCY-22-GFP are weakly or not exported to AMsh, respectively. Therefore, export of ciliary material from ciliated neurons to the supporting glia in physiological conditions might differ accordingly to cargo and/or neuron properties (see more detail in *Author response to reviewers*).

What is the purpose of ciliary ectocytosis? Cilia size, composition, and signaling are tightly regulated (*Carter and Blacque, 2019*). For example, tuning of the Hedgehog signaling is achieved by controlling cilia entry, removal and disposal of Hedgehog effectors (*Rohatgi et al., 2007*; *Wang et al., 2009*; *Bangs and Anderson, 2017*). The best-known mechanisms for entry and removal of proteins from cilia are mediated by the IFT cargo adapters IFT-A and BBSome, respectively. Several other mechanisms are described to remove proteins from cilia including endocytosis at PCMC and ciliary ectocytosis (*Kaplan et al., 2012*; *Nager et al., 2017*). We observed that ciliary ectocytosis is a by-product of the strong polarized traffic of GCY-22-wrmScarlet towards the distal dendrite and cilia. Polarized traffic of GCY-22-wrmScarlet to the distal dendrite is maintained in *daf-19* mutants together with its export to AMsh despite the absence of ciliary structure. Polarized traffic of GCY-22-wrmScarlet to the cilia is lost in *unc-101(m1)* together with its export to AMsh despite the cilia maintenance in *unc-101(m1)*. As previously suggested, we hypothesize that ciliary ectocytosis represents one mechanism to balance out the continuous import of membrane and membrane proteins to cilia (*Carter and Blacque, 2019*). Our results suggest that trafficking imbalance within a cilia subcompartment can be reduced by local ectocytosis of cargoes. In *bbs-8* mutants where cargo retrieval is reduced, GCY-22-GFP or TSP-6-wrmScarlet cargoes should accumulate in cilia. Instead, we observe more EVs are released from cilia such that GCY-22-GFP fluorescence is reduced in the ASER cilium of *bbs-8* mutants. Accordingly, more EVs were shown to accumulate in the amphid sensilla lumen of *bbs-8(nx77)* mutants (*Akella et al., 2020*). Therefore, ectocytosis likely contributes to reduce cargo accumulation in cilia of *bbs-8(nx77)* mutants. Similarly, in IMCD3 kidney cell line, cilia defective for BBSome retrieval increased receptor removal by ectocytosis (*Nager et al., 2017*). In *osm-3(p802)* mutants, where entry and anterograde transport of cargoes in cilia are reduced, GCY-22-GFP or TSP-6-wrmScarlet cargoes accumulate in PCMC. Concomitantly, fourfold more EVs are released from *osm-3* PCMC to the supporting glia than

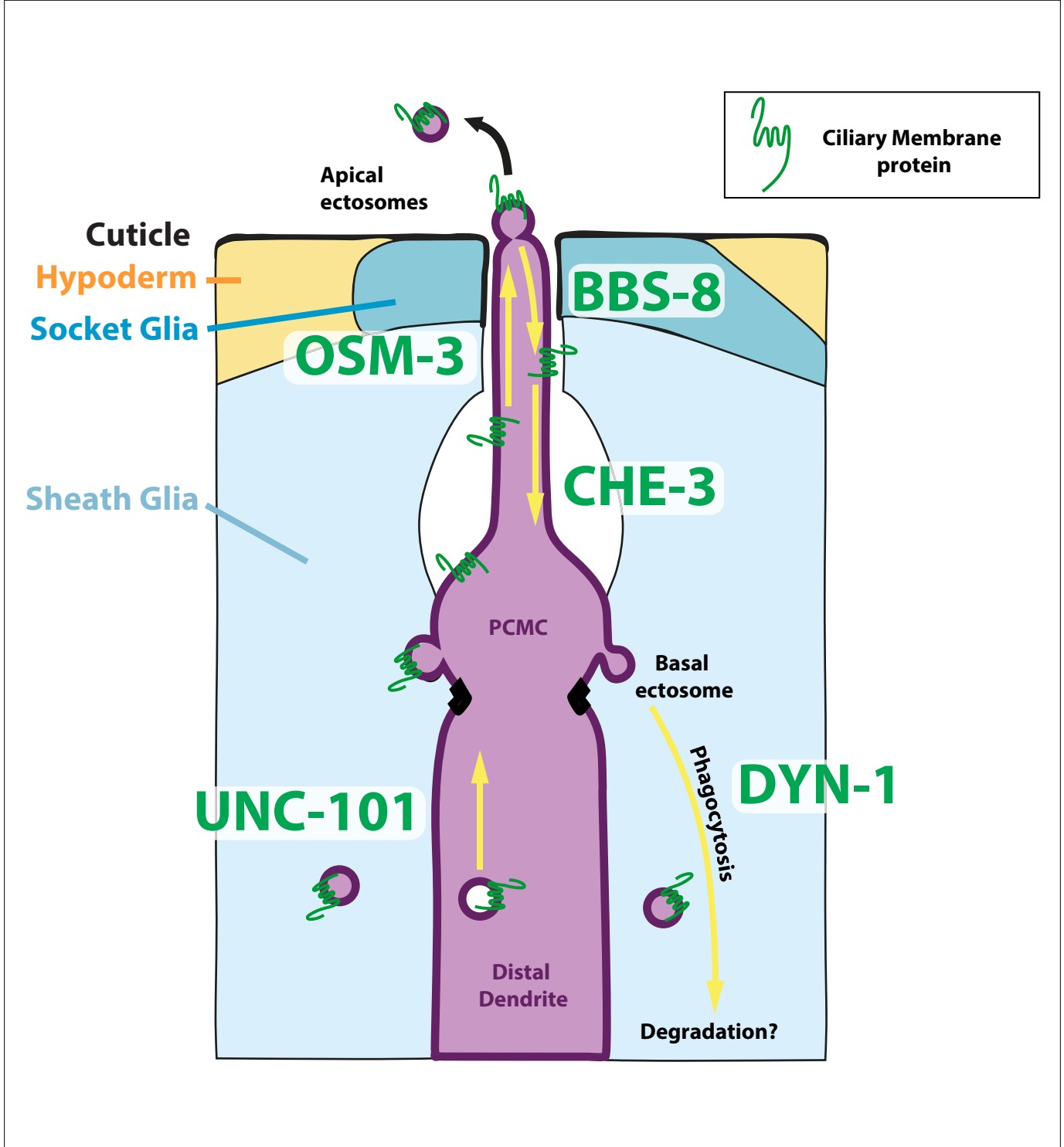

**Figure 9.** Mechanisms underlying ectocytosis from ciliated neurons. Ectocytosis occurs from two different ciliary locations: the cilia tip and the periciliary membrane compartment (PCMC). When ectosomes are shed from the cilia tip, they are environmentally released. Sheath glia (light blue) embeds the ciliary base of ciliated neurons, when ectosomes bud from the base they are concomitantly phagocytosed by their associated sheath glia. Overexpression of ciliary membrane proteins or mutations in genes involved in cilia trafficking, like *osm-3, bbs-8,* and *che-3*, fail to balance import and removal of ciliary membrane proteins, leading to local accumulation of cargo in cilia proper and/or to PCMC. It results in increased ectocytosis events to the sheath glia in *osm-3* and *che-3* and/or to the outside in *bbs-8*. On the contrary, reduced cilia import of ciliary membrane protein in *unc-101* – involved in sorting cilia cargo from Trans Golgi Network – reduced ectocytosis. Besides its crucial role in secretion of extracellular matrix content, we suggest that glia also plays an important function to maintain cilia structure and composition and to recycle ectocytosed material. Cell-specific

*Figure 9 continued on next page*

*Figure 9 continued*

manipulations of AMsh glia phagocytic activity by expression of DYN-1 dominant negative transgene suggest that a tight regulation of sheath glia phagocytosis contributes to shape the nerve receptive endings. We suggest a model where apical ectocytosis is inherent to most or all ciliated neurons of *C. elegans* and where neurons and glia cooperate to readily remove basal ectosome from ciliary membranes when cargoes accumulate in PCMC (magenta).

in controls. Therefore, we suggest that both apical and basal ectocytosis provide a safeguard measure to maintain appropriate cilia composition.

Does ciliary ectocytosis occur in pathological conditions? Conditions leading to transient accumulation of cargo in one cilium subcompartment promote local ectocytosis from this subcompartment. Therefore, cargo ectocytosis might protect from cargo accumulation in ciliopathies, reducing the load in cilia and/or PCMC. One possibility to explain PCMC accumulation of cargoes and increased basal ectocytosis in overexpression strains would be a saturation of the IFT retrieval machinery by overexpression of ciliary membrane proteins. Interestingly, PCMC expansion and basal ectocytosis observed in *osm-3(p802)* and *che-3(cas511)* in knocked in GCY-22-GFP are reminiscent of PCMC expansion and basal ectocytosis observed in strains overexpressing GCY-22-wrmScarlet. This observation highlights the risk of artifactual localization of cargoes and artifactual EV release in cilia overexpressing cargo proteins.

How does cargo enter ectosomes? Ciliary ectosome release was shown to involve ciliary accumulation of PI(4,5)P2 and subsequent actin polymerization in cilia (*Phua et al., 2017*; *Nager et al., 2017*). However, how EV cargoes are sorted remain poorly understood. Interestingly, all overexpressed ciliary membrane proteins tested apart SRBC-64 can enter EV when overexpressed. Although ciliary ectosomes derive from cilia membrane, their protein composition is different (*Long et al., 2016*). Therefore, some bias for EV entry must exist. We show that sorting of the same cargo (GCY-22-GFP or TSP-6-wrmScarlet) to basal or distal ectosomes is affected by cilia trafficking biases in *bbs-8* and *osm-3* mutants. Therefore, cargo entry in basal and/or distal ectosomes does not only result from a selective sorting but rather results from their enrichment in PCMC or cilia tip when ectosome budding occurs. This model is supported by ectosome biogenesis time lapses. Sensory cilia of *C. elegans* vary in terms of axonemal structure, microtubule arrangements, post-translational modifications of tubulin, as well as IFT motors composition (*Perkins et al., 1986*; *O'Hagan et al., 2017*; *Silva et al., 2017*; *Doroquez, 2014*; *Akella and Barr, 2021*). These cilia specificities contribute to determine whether cargoes are enriched in distal or basal cilia in a given cell type. This can explain the high propensity of male ciliated neurons to produce apical ectosomes carrying PKD-2 released outside, rather than basal ectosome exporting PKD-2 to the associated glia. In addition to cargo trafficking bias, cargo accumulation can play an active role in EV biogenesis by their bending properties, their interactions with bending proteins, and/or their preferential sorting to bending membranes. For example, clustering of membrane proteins with intrinsic conical shape creates microdomains, which can bend their associated membranes (*Aimon et al., 2014*). Tetraspanins were previously suggested to contribute to EVs biogenesis and cargo sorting to the EVs (*Andreu and Yanez-Mo, 2014*). TSP-6 overexpression might therefore facilitate EV biogenesis by its effect on protein clustering, membrane curvature, and/or its sorting to high curvature regions of the plasma membrane as suggested for its mammalian ortholog, the tetraspanin CD9 (*Umeda et al., 2020*). In agreement to an active role for cargo in shaping plasma membrane, we show that EV composition correlates with changes in EV diameter. This observation suggests that cargoes interact within the plasma membrane to define the EVs size. For example, GCY-22 overexpression might neutralize the bending effects of TSP-6, leading to larger EVs. Once membranes start bending, curvature-dependent sorting was recently observed to sort activated GPCRs (*Rosholm et al., 2017*).

Does glia capture of EV contribute to maintain cilia shape and function? We show that EVs remain produced by AFD receptive endings in the absence of AMsh glia; these EVs were alternatively taken up by hypodermal and neuronal cells. Therefore, glial phagocytic activity is secondary to the inherent formation of EVs by NREs, rather than an active pruning process. Because the sheath glia embeds partially or totally the NREs, glial phagocytosis was expected to go hand in hand with ectocytosis events. When AMsh phagocytic activity is disturbed by *dyn-1* dominant negative, the NRE morphology of ASH, ASER, AWC, and AFD receptive endings was altered, including formation of filopodial-like protrusions originating from PCMC. Not surprisingly, disrupted cilia morphology

correlates with abnormal ASER and ASH sensory responses. We suggest that neurons and glia would therefore cooperate to readily remove basal ectosomes from PCMC (*Figure 9*). Interestingly, AWC wing cilia and AFD NRE do not collapse in animals expressing *dyn-1* dominant negative in AMsh. Also, AWC and AFD-associated sensory responses remain normal. These observations suggest that the microenvironment of embedded NREs is maintained, contrasting with mutants and transgenics altering AMsh secretome or reducing AMsh exocytosis (*Bacaj et al., 2008*; *Singhvi et al., 2016*; *Wallace et al., 2016*). Nevertheless, we cannot exclude *dyn-1* dominant negative could alter PCMC shape by mechanisms unrelated to EV endocytosis. Raiders et al. showed that CED-10 activity in AMsh dictates AFD microvilli engulfment rate and AFD microvilli shape (*Raiders et al., 2021*). Instead, we could not observe an effect of CED-10 activity on cilia cargo engulfment (see more details in *Author response to reviewers*).

Besides its crucial role in secretion of extracellular matrix to the sensory pores, we suggest that *C. elegans* sheath glia also plays an important function to maintain cilia structure and composition by capture and recycling of ectosome discarded by associated neurons. Similarly, large EVs called exophers are extruded by touch neurons and phagocytosed by the associated hypodermis (*Melentijevic et al., 2017*). In mammals, photoreceptor neurons coexist with closely apposed retinal pigmented epithelium (RPE), which phagocytoses a packet of ~100 discs shed daily from the cilia tip of each photoreceptor (*Nachury and Mick, 2019*). Mutations in MERKT, a transmembrane protein involved in recognition and phagocytosis of the photoreceptor outer segments by RPE, lead to accumulation of debris and result in loss of vision and photoreceptor degeneration (*D'Cruz et al., 2000*). While the export of ciliary cargoes to the outside or to sheath glia suggests ciliary EVs represent a form of cellular disposal, several examples of bioactive ciliary EVs carrying information across cells are reported (*Wood and Rosenbaum, 2015*). It is possible that ciliary ectosomes have additional significance in cell-to-cell communication between cilia and glia. The export of ciliary ectosome from neurons to glia in pathological conditions contributes to reduce proteostatic stress in cilia and might signal this stress to the supporting glia. Interestingly, the cilia-defective mutants *daf-19* and *bbs-8* also show defective sheath lumen morphogenesis (*Perens and Shaham, 2005*; *Akella et al., 2020*). This suggests an interesting possibility where neuronal ectosomes could signal the cilia state to AMsh, while the latter responds by secreting factors in the vicinity of neuronal receptive endings (*Akella et al., 2020*; *Mukhopadhyay et al., 2008*; *Wallace et al., 2016*).

We show that ectocytosis is constitutive of most – potentially all – *C. elegans* sensory cilia and contributes to grant appropriate cilia composition, structure, and function. Cilia trafficking works as the main mechanism to traffic proteins in an out of the cilia. Ectocytosis might act as a safeguard alternative to reroute accumulated material when cilia trafficking is disrupted. Many questions remain, such as the cargo sorting machinery or the potential function of ectosome export to glia. Here, we establish an in vivo model to address questions relative to ciliary ectosomes production and their potential role in a neuron-glia ensemble.

## Materials and methods

### Strain and genetics

*C. elegans* were cultured on NGM agar plates provided with *Escherichia coli* OP50 bacteria and grown under standard conditions unless otherwise indicated (*Brenner, 1974*; *Wood, 1988*). All strains were grown at 20 °C unless otherwise indicated. Strains used in this study are listed in *Supplementary file 1*.

### Molecular biology and transgenic strains

*C. elegans* N2 genomic DNA was used as template for cloning PCRs. Cloning PCRs were performed using Phusion High Fidelity DNA polymerase (M0530L, New England BioLabs) and then validated by Sanger sequencing. Cloning information is provided in *Supplementary file 1*. *C. elegans*-optimized wrmScarlet was a gift from Thomas Boulin (*El Mouridi et al., 2017*).

Plasmids used to generate transgenic animals were generated by means of Multisite Three-Fragment Gateway Cloning (Invitrogen, Thermo Fisher Scientific, MA). In brief, PCR fragments containing AttB recombination sequences were recombined into DONOR vectors by means of BP Clonase reactions (Invitrogen, Thermo Fisher Scientific), effectively creating a collection of ENTRY

clones. Three entry clones, (Pos1 + Pos2 + Pos3), and a destination vector, were used to create a pEXPRESSION vector. The destination vector used in the constructs for this study was modified from the pDEST R4-R3 Vector, a 3′UTR sequence of the *let-858* gene was added between attR3 and AmpR. Plasmids used in this study are available in *Supplementary file 1*.

Site-specific mutagenesis for DYN-1(K46A) dominant negative mutation was performed by standard PCR method using a plasmid with the wild-type *dyn-1* genomic sequence as template. Overlapping primers of ≈40 bp were designed and PCR reaction was set for 25 cycles using Pfu Turbo polymerase. Following the reaction, mixes were digested with DpnI to remove bacterial methylated DNA and transformed into DH5-alpha competent bacteria. Plasmids DNA was extracted, and mutations were confirmed by sequencing of plasmid DNA at Eurofins Genomics (Ebersberg, Germany). All transgenic worms were generated by microinjection with standard techniques (*Mello et al., 1991*). For most injected constructs, injection mixes were composed of 30 ng/µl targeting constructs, 30 ng/µl of co-injection markers, and 40 ng/µl of 1 kb Plus DNA mass ladder (Invitrogen, Thermo Fisher Scientific) as carrier DNA to have a final injection mix concentration of 100 ng/µl. For *tsp-6* constructs, a final concentration of 5 ng/µl of targeting construct was used. For mKate constructs, a final concentration of 15 ng/µl. Each promoter used was first tested for expression pattern in order to make sure it was only expressed in neurons and never in glial cells. Due to mosaicism generated by injection of unstable extrachromosomal arrays, three independent transgenic lines were generated for each of the constructs used in this study, only one strain was shown to be representative of the phenotype observed in the three independent transgenic lines.

### TSP-6 CRISPR/Cas9 knock-in

Strain PLT03 (tsp-6::wrmScarlet) was generated by SunyBiotech Corporation corresponding to PHX4122 t*sp-6(syb4122)*. A wrmScarlet cassette was introduced c-terminal to *tsp-6* using the repair template: **AGCTGCTTGCGATGATATTCTCTTGTATCATTATTGGGGCCGTAAAGGAGAAACGCTC-CCAA GC T-[wrmScarlet sequence]-TAGATAATTCAATTGGTCTTTGTACTTGTTTATGCTTGGCCG TGTTTCACGTTTTGGT**, where bold refers to modifications to the original sequence. The wrmScarlet insertion in PLT03 strain was later verified by Sanger sequencing.

### DiI staining

Animals were synchronized by selecting L4 larva the day prior to the assay. Worms were stained using a modified version of Wormatlas Anatomical Methods (*Hall and Altun, 2008*). We used the lipophilic DiI (1,1′-dioctadecyl-3,3,3',3'-tetramethylindo carbocyanine perchlorate) (#42364, Sigma-Aldrich) to stain a subset of amphid sensory neurons (ASK, ADL, ASI, AWB, ASH, and ASJ). Briefly, worms were washed from plates with M9 and placed on a 1.5 ml microcentrifuge tubes. Worms were washed twice with M9 afterwards by spinning them down at 3000 rpms to bring the worms to the bottom of the tube. Worms were treated with M9 or with M9 supplemented with 25 mM sodium azide (NaN$_3$) (#S2002, Sigma-Aldrich, MO) 15 min prior to staining (see *Figure 1—figure supplement 1A*). 15 min of 25 mM NaN$_3$ was sufficient to fully anesthetize the animals as evaluated by the cease of pharyngeal pumping. DiI (2 mg/ml) stock solution was diluted 1:200 for staining. For staining, the worms were incubated for 20 min in the dark on a rocker at 30 rpms in M9 or in M9 + 25 mM NaN$_3$. After staining, worms were either washed three times with M9 + 25 mM NaN$_3$ and directly mounted on a microscope slide for imaging or were left on OP50-seeded plates and allowed to recover from anesthesia for 1 hr. After 1 hr of NaN$_3$ removal (sodium azide washout), the recovered animals were directly mounted for imaging.

### Image acquisition

Worms were synchronized either by bleaching a population of gravid worms or by an egg-laying window. Worms were reared at 20 °C up to the right stage. All animals were imaged at day 1 adulthood unless otherwise stated. Synchronized animals were mounted on 2% agarose pads and anesthetized with 25 mM NaN$_3$ dissolved in M9 solution. Images were acquired in the following 10–60 min after animals were anesthetized. Images were acquired at the Light Microscopy Facility LiMiF (http://limif.ulb.ac.be) at the Université Libre de Bruxelles, Faculté de Médecine, Campus Erasme, on a LSM780NLO confocal system fitted on an Observer Z1 inverted microscope (Carl Zeiss, Oberkochen, Germany). Images in which the full animal's head is displayed were acquired using a LD C-Apochromat

40× /1.1 W Korr M27 objective. The settings for these images were as follows: frame size was set at 1024 × 1024 pixels with a pixel size of 0.13 μm × 0.13 μm, pinhole size was set to 1 Airy Unit, Z-step optical sections varied across images (from 0.3 to 0.64 μm/step size) depending on the desired ROI volume (ranging from 10 to 35 μm in the Z axis), pixel dwell was set to 0.79 μs, and averaging was set to 4. High-resolution details (inset images) were acquired using an alpha Plan Apochromat 63× /1.46 Oil Korr M27 objective. The settings for these images were as follows: frame size was set to 412 × 412 pixels with a pixel size of 0.08 μm × 0.08 μm, pinhole size was set to 1 Airy Unit, Z-step optical sections varied across images (from 0.2 to 0.3 μm/step size) depending on the desired ROI volume (ranging from 10 to 20 μm in the Z axis), pixel dwell was set to 1.95 μs, and averaging was set to 4. Images with a single channel were acquired using GaAsP detector. Images with multiple channels were acquired simultaneously using PMT detector for the following fluorophores: CFP or mEGFP, and the GaAsP detector for wrmScarlet and mCherry signals. For individual channel acquisitions, the Main Beam Splitter matched the excitation wavelength of each used fluorophore. For simultaneous CFP/wrmScarlet or mKate images, a 458/543 Main Beam Splitter was used. For simultaneous mEGFP/wrmScarlet or mCherry, a 488/543 Main Beam Splitter was used. The following fluorophores excitation (Ex) and detection wavelengths (DW) were used: CFP (Ex: 458 nm – DW: 463–558 nm), mEGFP (Ex: 488 nm – DW: 493–569 nm), wrmScarlet/mKate/DiI (Ex: 543 nm – DW: 570–695 nm), and mCherry (Ex: 543 nm – DW: 588–695 nm). Fluorophore excitation and detection wavelength ranges were set according to the information available FPbase database for each fluorophore (*Lambert, 2019*). Laser power and detector gain settings were adjusted to maximize signal-to-noise ratio and minimize saturation when possible. Images were saved in .czi Zeiss file format.

Images used to quantify the number of GCY-22 ectosomes within AMsh were done using an Axioimager Z1 microscope. Images of the full animal's head were acquired using a Plan Apochromat 20× /0.8 M27 objective and an AxioCam MR R3 camera (Carl Zeiss,). The settings for these images were as follows: frame size was set at 1388 × 1040 pixels with a pixel size of 0.323 μm × 0.323 μm. For CFP/mEGFP channel, excitation was done using a 488 nm lamp and using the following filters for excitation and wavelength detection (excitation filter: 450–490 nm/detection filter: 500–550 nm), and exposure time was set to 20 ms. For wrmScarlet channel, excitation was done using a 453 nm lamp and using the following filters for excitation and wavelength detection (excitation filter: 538–562 nm/detection filter: 570–640 nm), and exposure time was set to 600 ms.

## In vivo time-lapse imaging

Animals were synchronized as mentioned before and imaged at day 1 adulthood. For time-lapse imaging, a drop of anesthetic solution: 10 mM tetramisole hydrochloride (#L9756, Sigma-Aldrich, St. Louis, MO) in M9 was placed in a FluoroDish (FD35-100, World Precision Instrument, Inc, FL). Several animals were placed in the drop for 10–15 min prior to imaging until anesthetized. Immobilized worms were then covered with a layer of 4% agarose and maintained in anesthetic solution throughout the acquisition duration. We used the LSM780NLO confocal microscope with either the LD C-Apochromat 40× /1.1 W Korr M27 objective or the alpha Plan Apochromat 63× /1.46 Oil Korr M27 objective to acquire the time-series images. Time intervals between frames were set as follows for each video: *Video 1* (1.23 s), *Videos 2–4* (940 ms), *Video 5* (2.84 s), *Video 6* (1.27 s), *Video 7* (2 or 5 s), *Video 8* (2 s), and *Video 9* (2.84 s). A single-focal plane representative of the NRE area was used to acquire the time series using a pinhole size of 1 AU. For *Videos 6 and 9*, pinhole size was set to 2 AU to maximize signal. Acquisition settings were set as described previously.

## Image processing

Confocal images were processed using FIJI (*Schindelin et al., 2012*) and Zen 2.6 Pro (Blue edition) software (Carl Zeiss). Z-stack acquisitions were converted into a 2D image using maximum intensity projections to obtain a flattened image representative of the 3D volume. Time-series images were processed in FIJI. The FIJI plugin StackReg (*Thevenaz et al., 1998*) was used to correct for slight XY animal movements occurring during acquisition (using rigid body as correction option). Time series were finally ensembled using Kapwing Studio online tool (Kapwing Resources, San Francisco, CA).

To obtain fluorescence intensity, different ROIs were drawn for cilia, PCMC, distal dendrite neuron cell body, or AMsh cell body. The fluorescence of each ROI was calculated. Background fluorescence was subtracted using the following formula: (CTFC = Integrated Density - Area of selected cell × Mean

fluorescence of background readings). For glia/neuron fluorescence ratio, we calculated obtaining the CTFC fluorescence values as stated above. Then the ROI's CTFC values in AMsh were divided by CTFC values in neurons to obtain a glia/neuron fluorescence ratio.

To quantify the export of GCY-22-wrmScarlet from ASER, all fluorescent vesicles located within AMsh cell body were manually counted on screen across the whole AMsh volume. CFP/mEGFP channel images were used to identify the limits of the AMsh cell body.

To quantify neuron NRE morphologies, maximum intensity projections were analyzed to quantify cilia length (measured from the cilia tip to the enlargement at PCMC), for PCMC area (excluding the cilium proper, up to the distal dendrite, based on the enriched GCY-22-wrmScarlet). ASER and ASH morphological classification is based on the number of tubulated (diameter <1 µm) and non-tubulated diverticula (not showing any sign of pinch) and originating from a constant 4 µm$^2$ PCMC. AWC morphological classification is based on the presence or absence of wing membranous expansions and the presence of ectopic cilia branching. AFD morphological classification is based on the number and length of AFD microvilli (length of microvilli in N2 animals was considered to be the baseline) and on the shape of AFD PCMC (protruding base bulges, extensions, and branching).

Vesicle size was calculated using Zen software line measurement tool, a line was traced from side to side at the midline of each vesicle, obtaining an approximate value of each vesicle's diameter. To analyze the vesicle content in the animals co-expressing of TSP-6-wrmScarlet and GCY-22-mEGFP in ASER neuron, maximum intensity projections of images of a predetermined size were analyzed to determine the number of vesicles containing TSP-6-wrmScarlet alone, GCY-22-mEGFP alone, and vesicles containing both TSP-6/GCY-22; percentages were determined by dividing the number of vesicles of each group to the total number of vesicles.

To analyze co-localization of animals co-expressing TSP-6-wrmScarlet and GCY-22-mEGFP in ASER neuron, we used Imaris software version 9 (Oxford Instruments, Zurich, Switzerland). We first established a ROI containing the cilia and their base. Pearson's coefficient was calculated by automatic thresholding of background voxels in the selected ROI. The displayed coefficient was averaged across all the animals measured to obtain an estimate of fluorescence co-localization for both proteins.

To count the number of TSP-6/TSP-7-wrmScarlet carrying vesicles within AMsh cell soma, Z-stack confocal images of the AMsh region were first converted to 2D-projections, then a ROI was drawn outlining the glia by aid of the AMsh::CFP co-expression marker. We used the detect particles function from the ImageJ plugin ComDet v.0.5.5 to score the number of vesicles within this given ROI with the following input parameters: (Segmenting larger particles: Yes, Approximate particle size [Min]: 3.00 pix = 0.39 µm, Intensity Threshold [in SD]: 4.00).

To assess GCY-22 ciliary localization, linescan analysis was performed using ImageJ linescan tool. Images of the cilia region of animals expressing mKate in ASER neuron and endogenously tagged GCY-22::GFP were used to do this characterization. A line was traced from the distal dendrite to the most distal GFP signal, including EVs still attached to ASER cilium or already detached. Linescans were aligned on the presumed transition zone based on the local drop of signal for the mKate channel within cilium compared to PCMC. GFP and RFP signals were averaged across >21 animals and plotted with standard deviations.

## Chemotaxis assays

Prior to chemotaxis assays, animals were reared in standard NGM medium (containing ~50 mM NaCl) seeded with a *E. coli OP50* bacterial lawn, making sure animals were not starved and that plates were not overcrowded with worms. Chemotaxis assays were performed using day 1 adult *C. elegans*, and adults were obtained by synchronizing a population of animals by doing a 2 hr egg-laying window 72 hr prior to the assay. Chemotaxis assays were performed with 9 cm Petri dishes.

Chemotaxis plates containing soluble chemicals were prepared to generate linear gradients of attractant (NaCl) or repellent ($CuSO_4$) compounds across the length of the agar plate. 9 cm Petri dishes were used as a spatial container, and the gradient was prepared as follows: first, Petri dishes were elevated using an electroporation cuvette cap to tilt the plate at an approximated 5° angle, the plate was filled with ~15–20 ml of melted 2% agar solution containing 50 mM of NaCl or 5 mM of $CuSO_4$ (see *Figure 8—figure supplement 1A and B*). Once the first layer of solution had solidified, the plate was positioned on a flat surface and a second solution of melted 2% agar (<50 °C) containing no NaCl or $CuSO_4$ was poured on top. Plates were left open throughout the preparation to

allow the surface to dry, and the gradient was allowed to be established for approximately 1 hr prior to the assay. 30 animals were positioned in the center of each assay plate and allowed to navigate through the plate for 30 min. At this given time point, animal's position was marked in the plate. To score the chemotaxis index, the plate was divided into two scoring regions and the chemotaxis index was calculated following the formula described in *Figure 8—figure supplement 1C*.

Chemotaxis to odorants (IAA) was performed in 9 cm plates containing 20 ml of chemotaxis medium (2% agar, 1 M KPO$_4$ [pH 6.6], 1 M CaCl$_2$, and 1 M MgSO$_4$). Plates were left to dry 30 min prior to the assay. Briefly, plates were divided into four equal quadrants (two for control [C] and two for test compound [T]). 5 min prior to the assay, 1 µl of NaN$_3$ was placed in each of the C and T spots. Animals were washed in a solution of chemotaxis buffer (1 M KPO$_4$ [pH 6.6], 1 M CaCl$_2$, and 1 M MgSO$_4$) and 50 and 250 animals were placed at the center of the plate, Kim wipes were used to remove buffer excess. Right after, 1 µl of EtOH was placed in each of the control spots and 1 µl of IAA [10$^{-2}$] was placed in the test spots, lid was closed, and plates were inverted. Worms were allowed to explore the chemotaxis plate 60 min of chemotaxis, and the assay was stopped by adding few drops of chloroform to the plate lid to immediately stop navigation of unanesthetized animals. Scoring was done as follows: chemotaxis index (C.I.) was calculated as the number of animals in the two test quadrants minus the number of animals in the two control quadrants, divided by the total number of animals. Animals that did not exit the landing zone were not counted (see *Figure 8—figure supplement 1D*).

## Thermotaxis assay

Thermotaxis protocol was adapted from *Kimata et al., 2012*. In brief, animals were reared at room temperature. The night prior to the assay, animals were incubated at 24 °C. Thermotaxis plates were filled with a thin 8 ml layer of Ttx medium (2% agar, 0.3% NaCl, and 25 mM KPO$_4$ [pH 6.0]). For a 9 cm distance, the generated temperature gradient was set between 20°C and 24°C. Animals were washed with M9 buffer (maintained at 24 °C) two times and a pellet with >300 animals was placed in a line at the center of the plate, excess buffer was removed by using Kim wipes. Plates were placed with the right outermost region at 24 °C with glycerol underneath to allow better temperature conduction. Animals were left to explore in the thermotaxis gradient for 1 hr, and the assay was immediately stopped by placing several drops of chloroform on the plate lid. The number of worms per temperature section was scored. Sections were divided as follows: section 1 (20–21°C), section 2 (21–22°C), section 3 (22–23°C), and section 4 (22–23°C) (see *Figure 8—figure supplement 1E*).

## Optogenetic assay

Animals were grown from embryo to adulthood on OP50 supplemented or not with 200 µM of all-trans-retinal (TR) (Sigma, #R2500) in the dark. Transgenic L4 were selected the day prior to the assay and placed on fresh OP50 supplemented or not with 200 µM of all-trans-retinal (Sigma, #R2500). Blue light (480 nm, 15 mw/mm$^2$) was used to activate channelrhodopsin in animals treated or non-treated with TR. We recorded the reversal response (minimum 1–2 backward head swings) during the first 10 s of light exposure.

## Ammonium chloride (NH$_4$Cl) treatment

Synchronized plates with L4 animals were washed and incubated in falcon tubes with M9 buffer supplemented with OP50 bacteria and 10 µm NH$_4$Cl. Falcon tubes shaken at 150 rpms O/N and worms were imaged the day after.

## Statistical analysis

Statistical analyses were performed using GraphPad Prism version 8.4.0 (San Diego, CA). For comparison between two groups with normally distributed data but unequal SD, unpaired t-test with Welch's correction was done. For comparison between multiple groups (>3), normally distributed data, and assuming equal standard deviations, one-way ANOVA was performed followed by Dunnett's post-hoc test to correct for multiple comparisons. Multiple comparisons were made between controls and the different experimental conditions, unless otherwise stated. For comparison between multiple groups (>3), normally distributed data but not assuming equal standard deviations, Brown–Forsythe ANOVA was performed followed by Dunnett's T3 post-hoc test to correct for multiple comparisons. Multiple comparisons were made between controls and the different

experimental conditions. For comparison between multiple groups (>3) in non-normally distributed data, nonparametric Kruskal–Wallis ANOVA was performed followed by Dunn's post-hoc test for multiple comparisons. For comparison between multiple groups comparing two independent variables, a two-way ANOVA was performed followed by Sidak's post-hoc test to correct for multiple comparisons. Experiment statistics appear in figure legends. Sample size is indicated in figures on top of each experimental group. In all performed tests, statistical significance threshold was set to $\alpha = 0.05$.

## Acknowledgements

Several strains were provided by the CGC, which is funded by the NIH Office of Research Infrastructure Programs (P40 OD010440). We acknowledge Renaud Legouis and members of our lab for their input on the manuscript, Junior Badziak for LaTeX support, and Teresa Lobo for technical help. We thank William Schaffer lab for AQ2335 strain, Geert Jansen for the GCY-22-GFP knock-in strain, Chiou Fen-Chuang for the AWC::DsRedII strain, and Thomas Boulin lab for wrmScarlet plasmid. We thank Jean-Marie Vanderwinden and the ULB Imaging Facility (LiMiF) for imaging advice.

## Additional information

### Funding

| Funder | Grant reference number | Author |
|---|---|---|
| Fonds De La Recherche Scientifique - FNRS | 22445636 | Patrick Laurent |
| Fonds De La Recherche Scientifique - FNRS | 5125519F | Adria Razzauti |

The funders had no role in study design, data collection and interpretation, or the decision to submit the work for publication.

### Author contributions

Adria Razzauti, Conceptualization, Formal analysis, Investigation, Methodology, Writing – original draft, Writing – review and editing; Patrick Laurent, Conceptualization, Funding acquisition, Methodology, Supervision, Writing – original draft, Writing – review and editing

### Author ORCIDs

Adria Razzauti ◉ http://orcid.org/0000-0002-1899-8599
Patrick Laurent ◉ http://orcid.org/0000-0001-5360-5597

### Decision letter and Author response

Decision letter https://doi.org/10.7554/eLife.67670.sa1
Author response https://doi.org/10.7554/eLife.67670.sa2

## Additional files

### Supplementary files

• Supplementary file 1. List of strains, plasmids, and primers used in this study.

• Transparent reporting form

• Source data 1. Source data related to this study.

### Data availability

All data generated or analysed during this study are included in the manuscript and supporting files. The manuscript is a microscopy study, all datapoints are represented in figures and figure supplements. Supplementary file 1 contains all material used in this work.

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
