## [Decision Letter]

**Acceptance summary:**

Here, Razzauti and Laurent show that membrane proteins in the cilia of *C. elegans* sensory neurons can be released in extracellular vesicles (EVs) produced by two distinct regions of the cilium. EVs released from one of these sites can be taken up by an adjacent glial cell, though the functional significance of this remains speculative. Importantly, the authors find that EV release is dramatically enhanced when ciliary membrane proteins are overexpressed, suggesting that this process acts as a safety valve that protects cilium integrity.

**Decision letter after peer review:**

Thank you for submitting your article "Ectosome uptake by glia sculpts *Caenorhabditis elegans* sensory cilia" for consideration by *eLife*. Your article has been reviewed by 2 peer reviewers, and the evaluation has been overseen by Doug Portman as the Reviewing Editor and Piali Sengupta as the Senior Editor. The following individual involved in review of your submission has agreed to reveal their identity: Maxence V Nachury (Reviewer #1).

The reviewers and editors have discussed their reviews with one another, and the Reviewing Editor has drafted this to help you prepare a revised submission.

We feel that your paper reports a number of interesting findings that advance our understanding of the biology of cilia and glia in *C. elegans*. However, there are two areas in which additional experiments would be necessary in a revised submission. In addition, each reviewer has provided additional comments and suggestions that we ask you to consider as you prepare a resubmission.

Essential revisions:

(1) Concerns about overexpression artifacts. Aside from the DiI experiments, all evidence for EV release and glial uptake relies on the overexpression of cilium proteins. This leads to questions about the extent to which your findings reflect typical neuronal/glial functions, or, rather, are observed only when cilia are disrupted by protein overexpression. The best way to address this would be to carry out additional experiments using an endogenously-tagged cargo protein. As pointed out by Reviewer #1 below, such a reagent has been published (GCY-22), so you would not necessarily need to generate any new transgenes.

(2) Concerns about interpretation of the experiments in which AMsh function is disrupted. As reviewer #2 notes, expression of dyn-1(dn) in AMsh, or loss of ced-10, might compromise AMsh function in ways that affect additional functions besides its ability to take up ciliary EVs. For this reason, it cannot be definitively concluded that the changes you see in NRE morphology and behavior in AMsh::dyn-1(dn) or ced-10 mutants are specific consequences of problems with EV uptake. Further, your experiments do not definitively show that EV uptake relies on the cell-autonomous function of ced-10 in AMsh, so these phenotypes could be indirect. These issues can be addressed with the following experiments.

(a) examine AMsh morphology in AMsh::dyn-10(dn) and/or ced-10 and AMsh::ced-10 animals. If AMsh morphology is grossly disrupted, this would indicate that its function may be more generally disrupted.

(b) similarly, examine AWC NRE morphology in these animals. Even if AMsh morphology is intact, its function can be generally compromised. In these cases, the AWC cilium typically collapses.

(c) carry out ced-10; AMsh::ced-10 rescue experiments, examining EV uptake, ASER NRE morphology, and salt chemotaxis. This would provide a second line of evidence that links EV uptake by AMsh to maintenance of cilium morphology and neuronal function.

*Reviewer #1 (Recommendations for the authors):*

Are the EVs that end up in glial cells ectosomes? Do they originate from the cilium proper or from the plasma membrane or are they exosomes? The intro (line 63-67) quotes Barr lab papers showing that mutation of MVB biogenesis genes STAM (ESCRT-0), MVB12 (ESCRT-I) or ALIX does not affect production of PKD2 EV by CEM neurons. Are ILVs absent from the MVB in these mutants? The point being raised is whether the hypothesis of MVB origin has been satisfactorily rejected. The question becomes relevant when one considers that the tetraspanins chosen by the authors are known markers of exosomes. In particular CD63 is a marker of exosomes but not of ectosomes as CD63 is enriched in ILVs of late endosomes but not present at the plasma membrane. Leaving open the possibility that the material is transferred via exosomes would be wise.

The most direct evidence for ectosomes budding from a ciliated neuron and engulfed by a support cell is presented in Video 6. In this video, where is the cilium and where are the microvilli? Can one determine the origin of the EV?

The use of overexpression system for the ciliary signaling receptors results in their localization to both cilia and PCMC and the dilation of the PCMC. It is therefore conceivable that the observed transfer of material from ciliated neuron to support cell is caused by receptor overexpression. The strong signal at the PCMC for all membrane proteins tested (except SRBC-64) leads one to wonder whether excess signaling receptors may accumulate at the PCMC and become shed. Figure 3C shows that SRBC-64 is present at similar levels in cilia and at the PCMC, unlike the other markers used in the study that are more highly enriched in the PCMC than in cilia. The fact that SRBC-64 is the only receptor that does not end in vesicles inside the support cell casts doubt on the ciliary origin of the EVs. The fluorescent signals also appear to be the weakest for this protein, suggesting that SRBC64 is the least overexpressed of all proteins tested. The result that EV transfer from ciliated neuron to support cell is increased when cilia are absent strengthen the interpretation that the material that is transferred is not of ciliary origin.

The expression of dominant negative dynamin in AMsh is interesting. Filopodial-looking extensions originating from the ASER are now detected more frequently. This suggests that these filopodial extensions from the ciliated neuron are phagocytosed by the AMsh. The images shown suggest that the filopodial extension do not originate from the cilium itself but rather from the PCMC or the distal dendrite. The strong reduction in TSP7 transfer from AFD to AMsh in mutant that lacks actin-base protrusion suggests that actin-based protrusion are the relevant entities for material sending by the donor cell.

It would be good to discuss the model that actin protrusions emanating from the PCMC are responsible for the material transfer from neurons to glial cells. Given the strong supportive evidence for actin protrusions playing a role in the process under study, these protrusions could be included in the model in Figure 8. The model should also show PCMC and cilia proper.

*Reviewer #2 (Recommendations for the authors):*

1. Release of EVs from sensory neurons and their uptake by surrounding cells has been previously described and should be cited (Melentijevic et al., Nature 2017).

2. The statement that "DiI passively diffuses in lipid membranes it contacts" is misleading. Several externally exposed neurons do not stain with DiI.

3. The statement that "most of the glial cells (46 out of 50) associate with groups of ciliated neurons to form sensory organs named sensilla" is confusing. All 50 glia do this.

4. The statement that the authors observed "barely detectable basal ectosomes… suggesting [their] size [is] close to the light diffraction limit of conventional confocal microscopes" is confusing. The light diffraction limit is related to resolution, that is, the ability to distinguish two objects as being distinct from each other. The authors seem to be referring to brightness which is not necessarily related to size.

5. The word "length" is misspelled in all figures where it appears.

Only as a note to the authors: I find the pseudo-typeset formatting to be very difficult to review. I would greatly prefer a simple text document with full-page figures and legends at the end. I understand that others may feel differently.

---

## [Author Response]

Essential revisions:(1) Concerns about overexpression artifacts. Aside from the DiI experiments, all evidence for EV release and glial uptake relies on the overexpression of cilium proteins. This leads to questions about the extent to which your findings reflect typical neuronal/glial functions, or, rather, are observed only when cilia are disrupted by protein overexpression. The best way to address this would be to carry out additional experiments using an endogenously-tagged cargo protein. As pointed out by Reviewer #1 below, such a reagent has been published (GCY-22), so you would not necessarily need to generate any new transgenes.

We agree with this concern. To address it, we carried out new experiments using 2 endogenously tagged EV cargos: GCY-22-GFP and TSP-6-wrmScarlet, both generated by CRISPR-Cas9 knock-in (GCY-22-GFP made by van der Burght et al., 2020 and TSP-6-wrmScarlet by us). In short, our results support these two cargo are released from cilia by EVs in absence of overexpression. However, this export is strongly reduced in knocked in strains compared to overexpression strains. Overexpression strains induce PCMC accumulation of cargo and EV export to the supporting glia. Similarly, *osm-3* and *che-3* cilia trafficking mutants induce PCMC accumulation of cargo and EV export to the supporting glia. Therefore, we suggest PCMC accumulation of cargo in overexpression strain leads to strong export of cargo to the supporting glia. PCMC accumulation of cargo might be explained by saturation of the ciliary trafficking machinery (Figures 4 and 5). We highlight the risk of artefacts when overexpressing cilia cargo in the discussion.

(2) Concerns about interpretation of the experiments in which AMsh function is disrupted. As reviewer #2 notes, expression of dyn-1(dn) in AMsh, or loss of ced-10, might compromise AMsh function in ways that affect additional functions besides its ability to take up ciliary EVs. For this reason, it cannot be definitively concluded that the changes you see in NRE morphology and behavior in AMsh::dyn-1(dn) or ced-10 mutants are specific consequences of problems with EV uptake. Further, your experiments do not definitively show that EV uptake relies on the cell-autonomous function of ced-10 in AMsh, so these phenotypes could be indirect. These issues can be addressed with the following experiments.(a) examine AMsh morphology in AMsh::dyn-10(dn) and/or ced-10 and AMsh::ced-10 animals. If AMsh morphology is grossly disrupted, this would indicate that its function may be more generally disrupted.(b) similarly, examine AWC NRE morphology in these animals. Even if AMsh morphology is intact, its function can be generally compromised. In these cases, the AWC cilium typically collapses.(c) carry out ced-10; AMsh::ced-10 rescue experiments, examining EV uptake, ASER NRE morphology, and salt chemotaxis. This would provide a second line of evidence that links EV uptake by AMsh to maintenance of cilium morphology and neuronal function.

We agree with these concerns. We tried to address them as much as possible. In short: both *ced10(n3246),* and AMsh::DYN-1(K46A) affect AMsh shape and position suggesting both manipulation alter AMsh cell biology. In addition, the position and shape of some neurons -likely ASI and ASJ- were also altered in *ced-10(n3246)* (Author response image 1) . Because our previous result showing a reduced EV capture by AMsh in *ced-10(n3246)* was not confirmed in further replicated experiments, we focused our additional experiments on the effects of AMsh::DYN-1(K46A).

**Author response image 1. sa2fig1:** (**A**) DiI filling experiments were performed for controls *and ced-10(n3246)*. *ced-10(n3246)* did not affect neuronal uptake of DiI nor the DiI export from ciliated neurons to AMsh glia. AMsh cell body position (dashed line is aligned to the position of the head ganglia) and projections (green arrows) are modified in *ced-10(n3246)* mutants. In *ced-10(n3246)*, neurons identified as ASI and ASJ are displaced posteriorly and/or show abnormal neurite extensions (**B**) The AMsh cell body position relative to the nose tip of the animals was measured for AMsh::DYN-1(K46A) and for *ced-10(n3246).* (**C**) We quantified the number of GCY-22-wrmScarlet carrying vesicles within AMsh in animals overexpressing GCY-22-wrmScarlet in ASER. Their number was increased in animals expressing AMsh::DYN-1(K46A) compared to controls. *ced-10(n3246) and alx-1(gk338)* did not modify the number of GCY-22-wrmScarlet carrying vesicles in AMsh.

These additional experiments lead to several conclusions: (1) The shape and position of AMsh are modified by AMsh::DYN-1(K46A), suggesting DYN-1 dominant negative transgene modifies the cell biology of AMsh. (2) In AMsh::DYN-1(K46A), ciliated ends of ASH, ASER, AWC and the microvilli of AFD are abnormally shaped. Interestingly it includes the production of filopodia-like protrusions originating from their PCMC. (3) ~10% of AFD and none of the AWC neurons receptive endings are truncated by expression of AMsh::DYN-1(K46A). (4) Sensory responses mediated by ASER and ASH are strongly reduced while sensory responses mediated by AFD and AWC are unaltered (Figures 7 and 8).

The absence of sensory defect in AWC and AFD-associated sensory response suggest the microenvironment of these two embedded NREs is maintained. This observation and absence of collapsed NRE strongly contrasts with the effects of AMsh ablation, AMsh exocytosis defect (by RAB-1(S25N) expression) or AMsh secretome defect (in *pros-1* mutants) on AWC and AFD (Bacaj et al., 2008; Singhvi et al., 2016; Wallace et al., 2016).

Altogether, we conclude that expression of dynamin dominant negative in AMsh alter cilia shape of all amphid neurons tested leading to sensory dysfunction in ASER and ASH. This effect can be explained by deficient EV uptake or by an indirect effect of dynamin dysfunction on AMsh function. We agree we cannot exclude DYN-1 dominant negative would compromise AMsh function in ways that affect additional functions besides its ability to take up ciliary EVs. We modified our discussion accordingly.

Reviewer #1 (Recommendations for the authors):Are the EVs that end up in glial cells ectosomes? Do they originate from the cilium proper or from the plasma membrane or are they exosomes? The intro (line 63-67) quotes Barr lab papers showing that mutation of MVB biogenesis genes STAM (ESCRT-0), MVB12 (ESCRT-I) or ALIX does not affect production of PKD2 EV by CEM neurons. Are ILVs absent from the MVB in these mutants? The point being raised is whether the hypothesis of MVB origin has been satisfactorily rejected. The question becomes relevant when one considers that the tetraspanins chosen by the authors are known markers of exosomes. In particular CD63 is a marker of exosomes but not of ectosomes as CD63 is enriched in ILVs of late endosomes but not present at the plasma membrane. Leaving open the possibility that the material is transferred via exosomes would be wise.

Prior evidence suggested ciliary EVs produced from PCMC of CEM correspond to ectosomes: omega shaped structures were observed at the PCMC of CEM neurons (Silva et al., 2017; Wang et al., 2014). Entry of plasma membrane cargo into ILV/exosomes require the ESCRT function (Hessvik and Llorente, 2018). Mutants for key ESCRT genes *stam-1* (ESCRT-0), *mvb-12* (ESCRT-I) and *alx-1* (ALIX) did not affect release of apical EVs carrying PKD-2-GFP (Wang et al., 2014).

In Video 6, we show GCY-22-wrmScarlet ectosomes budding from ASER PCMC to be captured by AMsh. We tested the effect of *alx-1* mutation on GCY-22-wrmScarlet export to AMsh and did not observe significant effects on its export to AMsh, suggesting basal release do not require MVB maturation (See first Figure in this Letter). Despite the described absence of MVB from cilia, we observed overexpressed TSP-7-wrmScarlet to be enriched in cilia of several *C. elegans* sensory neurons (Figure 2 —figure supplement 1). Finally, Videos 2 and 3 show the biogenesis of TSP-7wrmScarlet ectosomes budding from the cilia tip, not MVB fusion. Therefore, we believe TSP-7 behave as other ciliary membrane proteins.

We agree that CD63 is usually used as a marker of exosomes. However, CD63 also marks other EVs (Kowal et al., 2016) is present at plasma membrane (Pols and Klumperman, 2009). Mutations in the lysosome targeting motif of CD63 makes it to behave like CD9 (enrichement in large Ectosomes/EVs ;(Mathieu et al., 2020)) and this lysosome targeting motif is not present in TSP-7. Nevertheless, to avoid confusion, we did all new experiments with TSP-6 – the ortholog of CD9 – a marker of microvesicle/ectosomes. We leave open the possibility other cargos might be exported to AMsh through exosomes.

The most direct evidence for ectosomes budding from a ciliated neuron and engulfed by a support cell is presented in Video 6. In this video, where is the cilium and where are the microvilli? Can one determine the origin of the EV?

There might be a confusion between Video 6 showing a GCY-22-wrmScarlet ectosome budding from ASER PCMC and (current) Video 9 showing TSP-6-wrmScarlet microvilli budding EVs from the AFD microvilli. We improved the labelling and legends of Video 6 and Video 9 to avoid confusion.

The use of overexpression system for the ciliary signaling receptors results in their localization to both cilia and PCMC and the dilation of the PCMC. It is therefore conceivable that the observed transfer of material from ciliated neuron to support cell is caused by receptor overexpression. The strong signal at the PCMC for all membrane proteins tested (except SRBC-64) leads one to wonder whether excess signaling receptors may accumulate at the PCMC and become shed. Figure 3C shows that SRBC-64 is present at similar levels in cilia and at the PCMC, unlike the other markers used in the study that are more highly enriched in the PCMC than in cilia. The fact that SRBC-64 is the only receptor that does not end in vesicles inside the support cell casts doubt on the ciliary origin of the EVs. The fluorescent signals also appear to be the weakest for this protein, suggesting that SRBC64 is the least overexpressed of all proteins tested.

Reviewer #1 is right: Cilia receptors overexpression leads to their accumulation in PCMC and promotes their shedding to the glia. The PCMC enlargement and misshape caused by GCY-22wrmScarlet overexpression is similar to the PCMC enlargement and misshape caused by *osm-3* and *che-3* in GCY-22-GFP knock-in strain*.* Therefore, we suggest that – above a threshold – overexpression of cilia receptors might saturate the cilia trafficking machinery, leading to PCMC accumulation.

The result that EV transfer from ciliated neuron to support cell is increased when cilia are absent strengthen the interpretation that the material that is transferred is not of ciliary origin.The expression of dominant negative dynamin in AMsh is interesting. Filopodial-looking extensions originating from the ASER are now detected more frequently. This suggests that these filopodial extensions from the ciliated neuron are phagocytosed by the AMsh. The images shown suggest that the filopodial extension do not originate from the cilium itself but rather from the PCMC or the distal dendrite.

All experiment we display in Figure 7 are done with overexpression of cytoplasmic mKate which did not modify cilia shape and size in N2. In animals expressing DYN-1(DN) in AMsh, and mKate in neurons, filopodial-looking extensions are observed originating from the PCMC of ASH, ASER, AWC and AFD. We suggest that filopodial-looking extensions attached to PCMC might reflect ectosome undergoing abnormal phagocytosis from PCMC. This location might simply reflect where cilia closely contact AMsh glia membrane and/or specific properties of PCMC compared to cilia proper.

The strong reduction in TSP7 transfer from AFD to AMsh in mutant that lacks actin-base protrusion suggests that actin-based protrusion are the relevant entities for material sending by the donor cell. It would be good to discuss the model that actin protrusions emanating from the PCMC are responsible for the material transfer from neurons to glial cells.

Unfortunately, while increasing the N number the export reduction observed in *ced-10* vanished (See first Figure in this Letter). As an alternative, we tried to express CED-10(T17N) dominant negative in AMsh, this did not reduce EV export from ASER and we did not observe the filopodial-looking extensions. We cannot exclude a role for CED-10 and actin protrusions in PCMC; however, we did not explore this possibility yet.

Given the strong supportive evidence for actin protrusions playing a role in the process under study, these protrusions could be included in the model in Figure 8. The model should also show PCMC and cilia proper.

We now distinguish PCMC and cilia proper in the scheme in Figure 2B and in the model scheme in Figure 9.

Reviewer #2 (Recommendations for the authors):1. Release of EVs from sensory neurons and their uptake by surrounding cells has been previously described and should be cited (Melentijevic et al., Nature 2017).

Done.

2. The statement that "DiI passively diffuses in lipid membranes it contacts" is misleading. Several externally exposed neurons do not stain with DiI.

We modified the sentence.

3. The statement that "most of the glial cells (46 out of 50) associate with groups of ciliated neurons to form sensory organs named sensilla" is confusing. All 50 glia do this.

Thank you for this comment, we have now updated the introduction with the suggested change.

4. The statement that the authors observed "barely detectable basal ectosomes… suggesting [their] size [is] close to the light diffraction limit of conventional confocal microscopes" is confusing. The light diffraction limit is related to resolution, that is, the ability to distinguish two objects as being distinct from each other. The authors seem to be referring to brightness which is not necessarily related to size.

Reviewer #2 is correct, we apologize for this misunderstanding and removed this sentence in the manuscript.

5. The word "length" is misspelled in all figures where it appears.

Misspelled “Length” words have been changed appropriately.

References:

Akella JS, Barr MM. 2021. The tubulin code specializes neuronal cilia for extracellular vesicle release. Developmental Neurobiology 81:231–252. doi:10.1002/dneu.22787

Bacaj T, Tevlin M, Lu Y, Shaham S. 2008. Glia Are Essential for Sensory Organ Function in *C. elegans*. Science 322:744–747. doi:10.1126/science.1163074

Fazeli G, Trinkwalder M, Irmisch L, Wehman AM. 2016. *C. elegans* midbodies are released, phagocytosed and undergo LC3-dependent degradation independent of macroautophagy.

Journal of Cell Science 129:3721–3731. doi:10.1242/jcs.190223

Hessvik NP, Llorente A. 2018. Current knowledge on exosome biogenesis and release. Cell Mol Life Sci 75:193–208. doi:10.1007/s00018-017-2595-9

Kowal J, Arras G, Colombo M, Jouve M, Morath JP, Primdal-Bengtson B, Dingli F, Loew D, Tkach M, Théry C. 2016. Proteomic comparison defines novel markers to characterize heterogeneous populations of extracellular vesicle subtypes. PNAS 113:E968–E977.

doi:10.1073/pnas.1521230113

Mathieu M, Névo N, Jouve M, Valenzuela JI, Maurin M, Verweij F, Palmulli R, Lankar D, Dingli F, Loew D, Rubinstein E, Boncompain G, Perez F, Théry C. 2020. Specificities of exosome versus small ectosome secretion revealed by live intracellular tracking and synchronized extracellular vesicle release of CD9 and CD63. bioRxiv 2020.10.27.323766.

doi:10.1101/2020.10.27.323766

Pols MS, Klumperman J. 2009. Trafficking and function of the tetraspanin CD63. Experimental Cell Research, Special Review Issue on Intracellular Trafficking 315:1584–1592.

doi:10.1016/j.yexcr.2008.09.020

Silva M, Morsci N, Nguyen KCQ, Rizvi A, Rongo C, Hall DH, Barr MM. 2017. Cell-Specific alphaTubulin Isotype Regulates Ciliary Microtubule Ultrastructure, Intraflagellar Transport, and

Extracellular Vesicle Biology. Curr Biol. doi:10.1016/j.cub.2017.02.039

Singhvi A, Liu B, Friedman CJ, Fong J, Lu Y, Huang X-Y, Shaham S. 2016. A Glial K/Cl Transporter Controls Neuronal Receptive Ending Shape by Chloride Inhibition of an rGC. Cell 165:936–

948. doi:10.1016/j.cell.2016.03.026 van der Burght SN, Rademakers S, Johnson JL, Li C, Kremers GJ, Houtsmuller AB, Leroux MR,

Jansen G. 2020. Ciliary Tip Signaling Compartment Is Formed and Maintained by

Intraflagellar Transport. Curr Biol. doi:10.1016/j.cub.2020.08.032

Wallace SW, Singhvi A, Liang Y, Lu Y, Shaham S. 2016. PROS^-1^/Prospero Is a Major Regulator of the Glia-Specific Secretome Controlling Sensory-Neuron Shape and Function in *C. elegans*. Cell Reports 15:550–562. doi:10.1016/j.celrep.2016.03.051

Wang J, Silva M, Haas LA, Morsci NS, Nguyen KCQ, Hall DH, Barr MM. 2014. *C. elegans* Ciliated Sensory Neurons Release Extracellular Vesicles that Function in Animal Communication. Current Biology 24:519–525. doi:10.1016/j.cub.2014.01.002